# Toward stabilization of formamidinium lead iodide perovskites by defect control and composition engineering

Yuhang Liang [1,2] ✉, Feng Li [2] ✉, Xiangyuan Cui [3] ✉, Taoyuze Lv[2], Catherine Stampfl[2], Simon P. Ringer [3], Xudong Yang [4,5,6], Jun Huang [1] ✉ & Rongkun Zheng [2] ✉

Phase instability poses a serious challenge to the commercialization of formamidinium lead iodide ($FAPbI_3$)-based solar cells and optoelectronic devices. Here, we combine density functional theory and machine learning molecular dynamics simulations, to investigate the mechanism driving the undesired α-δ phase transition of $FAPbI_3$. Prevalent iodine vacancies and interstitials can significantly expedite the structural transition kinetics by inducing robust covalency during transition states. Extrinsically, the detrimental roles of atmospheric moisture and oxygen in degrading the $FAPbI_3$ perovskite phase are also rationalized. Significantly, we discover the compositional design principles by categorizing that A-site engineering primarily governs thermodynamics, whereas B-site doping can effectively manipulate the kinetics of the phase transition in $FAPbI_3$, highlighting lanthanide ions as promising B-site substitutes. A-B mixed doping emerges as an efficient strategy to synergistically stabilize α-$FAPbI_3$, as experimentally demonstrated by substantially higher initial optoelectronic characteristics and significantly enhanced phase stability in Cs-Eu doped $FAPbI_3$ as compared to its Cs-doped counterpart. This study provides scientific guidance for the design and optimization of long-term stable $FAPbI_3$-based solar cells and other optoelectronic devices through defect control and synergetic composition engineering.

Organic–inorganic halide perovskites have emerged as exceptional materials for next-generation photovoltaic applications, with solar power conversion efficiencies exceeding 25.8%[1,2], rivaling that of the well-established silicon-based solar cells. Currently, the high-efficiency perovskite solar cells are predominantly fabricated with formamidinium lead iodide ($FAPbI_3$), which exhibits higher thermal stability against decomposition into the secondary compound of $PbI_2$, a broader absorption spectrum that extends further into the red, and an ideal bandgap (-1.4 eV) closer to the Shockley–Queisser optimum, as compared with the prototypical methylammonium lead iodide ($MAPbI_3$)[3–5]. The superior optoelectronic properties of $FAPbI_3$ originate from the high-symmetry cubic perovskite structure (black α-phase)[6]. Regrettably, this phase is vulnerable and prone to transition into a photovoltaically-inactive yellow non-perovskite δ-phase with

[1]School of Chemical and Biomolecular Engineering, The University of Sydney, Sydney NSW 2006, Australia. [2]School of Physics, The University of Sydney, Sydney, NSW 2006, Australia. [3]School of Aerospace, Mechanical and Mechatronic Engineering, The University of Sydney, Sydney, NSW 2006, Australia. [4]State Key Laboratory of Metal Matrix Composites, Shanghai Jiao Tong University, Shanghai 200240, China. [5]Center of Hydrogen Science, School of Materials Science and Engineering, Shanghai Jiao Tong University, Shanghai 200240, China. [6]Zhangjiang Institute for Advanced Study, Shanghai Jiao Tong University, Shanghai 201210, China. ✉e-mail: yuhang.liang@sydney.edu.au; feng.li2@sydney.edu.au; carl.cui@sydney.edu.au; jun.huang@sydney.edu.au; rongkun.zheng@sydney.edu.au

hexagonal symmetry and a wide bandgap (~2.4 eV) below a temperature of 150 °C (refs. [4–11]), especially under humid environments or with excess photogenerated holes[12,13]. This instability is a major barrier to developing practical perovskite solar cells. An in-depth understanding of the key mechanisms underlying the phase instability of α-FAPbI$_3$ and the effective remedies are vital for achieving long-term stability in perovskite solar cells and other broadly ranged optoelectronic applications with high performance.

Traditionally, the studies of phase transitions of halide perovskites[14–16] have been mostly based on defect-free pristine models. The interaction between the phase transition and lattice defects has been typically overlooked, despite prevalent defects in metal halide perovskites especially synthesized through solution-based methods[17]. Indeed, it was reported that the presence of defects can induce local symmetry breaking and significantly deteriorate the mechanical stability of perovskites, facilitating lattice distortion and octahedral tilting[18–21]. We are thus confronted with fundamental questions: what are the roles of intrinsic defects or unintentional impurities in impacting the α-δ phase transition in FAPbI$_3$, and more importantly, how can the perovskite phase instability of FAPbI$_3$ be effectively remedied?

Efforts to stabilize the photovoltaically active FAPbI$_3$ lattice have led to various strategies, including composition engineering through doping engineering[4,6,7,11,22–28] and strain engineering[5,25,29]. The ternary composition of halide perovskites (ABX$_3$) provides a high degree of flexibility for doping engineering at its three available sites. Particularly, A–X doping for FAPbI$_3$ with MABr, MACl, CsBr, and/or CsCl has been one of the most popular strategies for suppressing the phase transition of FAPbI$_3$ (refs. [6,11,22,23,25,26]), although the underlying mechanism and effectiveness trend remain elusive. B-site doping engineering, despite being less explored due to higher binding energy[30], holds great potential as a breakthrough for enhancing perovskite stability, given the wider variety of dopant species[31,32]. Rare-earth elements, like lanthanide (Ln) ions, have been effective in improving the phase stability of traditional semiconductors[33] and the ambient stability of halide perovskites against thermal, moisture, and light conditions[34,35] by strengthening lattice chemical bonds. This approach could potentially stabilize α-FAPbI$_3$, even though further validation is needed. To further design stable and efficient photovoltaic and optoelectronic devices, a thorough and systematic exploration on how the doping of foreign cations at three available sites, especially with Ln ions at the B sites, impacts the phase transition thermodynamics and kinetics in FAPbI$_3$ is crucial (ref. [36]).

In this study, by performing comprehensive first-principles calculations combined with machine learning, we reveal that intrinsic defects can indeed play an important role in aggravating the instability of α-FAPbI$_3$. In particular, the presence of iodine defects, i.e., iodine vacancies and interstitials, can significantly lower the activation energy barrier of α-δ phase transition kinetics of FAPbI$_3$. We also identify the detrimental influence of atmospheric moisture and oxygen decomposition products on FAPbI$_3$ perovskite phase degradation. Our analysis shows that A-site cation engineering can effectively alter the thermodynamic driving force of the phase transition, while B-site doping plays a more significant role in affecting the phase transition kinetics, giving prominence to A-B mixed doping engineering as a highly efficient strategy for synergic improvements of stability and properties of FAPbI$_3$ perovskites. The analysis of the optimal the octahedral factor ($\mu$) range for the stable FA-based perovskite phase offers a quantitative basis for future Ln doping research to optimize halide perovskite materials. Furthermore, we provide direct experimental evidence of significantly improved optoelectronic properties and phase stability in Cs-Eu mixed-doped FAPbI$_3$ as compared to its Cs-doped counterpart. Our study provides in-depth insights into the thermodynamics and kinetics of α-δ phase transition in FAPbI$_3$ and establishes a basis for advanced device designs with "on-demand"

optoelectronic properties and stability through strategic defect and composition engineering.

## Results

### α-δ phase transition of FAPbI$_3$

Pure α-FAPbI$_3$ (black phase) features a cubic perovskite structure (ABX$_3$) with the sixfold coordinated PbI$_6$ inorganic octahedra connected at the corner and the larger FA organic cations in 12-fold coordination (Fig. 1 and Supplementary Fig. 1a). In contrast, δ-FAPbI$_3$ (yellow phase) exhibits a hexagonal structure with lower symmetry, where PbI$_6$ octahedra are face-sharing (Fig. 1 and Supplementary Fig. 1b). Electronically, α-FAPbI$_3$ has a favorable direct bandgap of 1.42 eV with highly dispersive bands near the valance band maximum (VBM) and conduction band minimum (CBM) (see Supplementary Fig. 1c, d), while the non-perovskite δ-FAPbI$_3$ exhibits a larger indirect bandgap of 2.53 eV with relatively flatter band edges (see Supplementary Fig. 1e, f), due to the disconnection of octahedra within the a–b plane that decreases the overlap of the Pb states and I states. Energetically, α-FAPbI$_3$ is higher in energy by 0.261 eV per formula unit than the δ-phase, which agrees with the experimental findings[4,8–10] of the metastable nature of the desired α-phase of FAPbI$_3$.

The undesired α−δ phase transition of FAPbI$_3$ active layers exhibits a large thermal hysteresis[10], indicative of a thermally activated process with a transition activation energy barrier. To quantitatively assess the underlying kinetics and gain a full understanding of phase transition mechanisms in FAPbI$_3$, we modeled the transition pathway of α-δ phase transition in FAPbI$_3$ using the well-established variable-cell nudged elastic band (VCNEB) method[37]. As shown in Fig. 1a, the α−δ phase transition undergoes an elongation of the a and b lattice parameters accompanied by a lattice contraction along the c direction. A volume expansion occurs at the transition state, followed by a significant volume contraction to form the final state of δ-FAPbI$_3$ (Fig. 1b). Similar behaviors were found in all-inorganic perovskite CsPbI$_3$ during its orthorhombic-to-hexagonal phase transition[14,38].

The phase transition involves complex atomic rearrangement especially on the inorganic octahedral skeleton, as shown in Fig. 1e–f. It starts with rotations of the octahedra, followed by breaking of Pb–I bonds in the a−b plane and forming a one-dimensional PbI$_4$ tetrahedral chain at the transition state. This process leads to a volume expansion and a sharp increase in the lattice cavity. As a result, the FA cations are separated with the FA-FA distance being enhanced by a factor of 1.28 as compared to the initial state. As the transition proceeds, the Pb ions of the PbI$_4$ tetrahedral chain move much closer, forming the new Pb–I bonds (as reflected by the sharp decrease of the c lattice after the transition state in Fig. 1a), and thus the face-shared PbI$_6$ octahedral chain, which further separates the chains within the a−b plane and enlarging the lattice cavity. This results in 1.09 times increase in FA-FA distance horizontally. Noted that in this process, the vertical compression effect is greater than the expansion in the horizontal direction, hence responsible for the reduction in the material's volume. As discussed above, the inorganic Pb–I lattice plays an important role in tailoring the band edges of the perovskites. A large bandgap fluctuation of FAPbI$_3$ (Fig. 1c) reflects the complex structural rearrangement of the inorganic skeleton during the phase transition. Moreover, the lattice variation during the transition implies that strain engineering could be a viable approach to remedy the phase instability of the FAPbI$_3$ active layers.

As shown in Fig. 1d, the α−δ phase transition of FAPbI$_3$ requires overcoming an activation energy barrier ($E_b$) of 0.622 eV/f.u., which is close to the previous results of 0.6–0.7 eV/f.u. via various computational methods[9,10,15,39]. Given the orientationally disordered nature of FA cations in the realistic perovskites[40], we further extracted four low-energy perovskite structures with different FA-cationic orientations from ab initio molecular dynamic (AIMD) simulations at room temperature, as shown in Supplementary Fig. 2a–d, and studied their

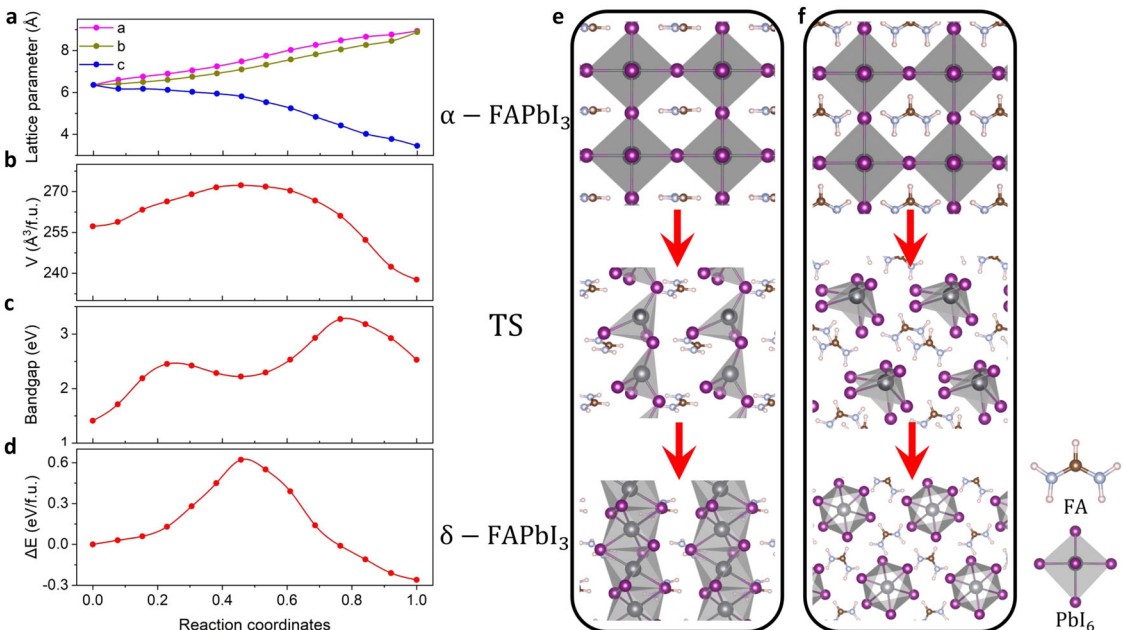

**Fig. 1 | α−δ phase transition in FAPbI₃.** Evolution of **a** the lattice parameters, **b** volume, **c** bandgap, and **d** the potential energy as a function of reaction coordinate of the α-δ phase transition in FAPbI₃, where the starting point ($x = 0$) of the reaction coordinate is defined as the configuration of the initial α-phase state, and the endpoint ($x = 1$) stands for the final δ-phase state configuration. **e** Side and **f** top views of the atomic displacements along the α−δ phase transition in FAPbI₃, where TS stands for the transition state.

phase transition properties. From the extracted Str.1 to Str.4, the orientational disorder of the FA cations is increasingly significant relative to the pristine model (Supplementary Fig. 1a). As shown in Supplementary Fig. 2e, these cation-disordered systems almost preserve the pristine phase transition barrier ($E_b \sim 0.62$ eV/f.u), whereas the energy difference between the α-phase and δ-phase ($\triangle E_{\delta-\alpha}$) of FAPbI₃ decreases. Notably, the reduction trend of $\triangle E_{\delta-\alpha}$ agrees with the degree of the FA-cationic disorder. This can be attributed to the fact that the rotation of A-site cations can thermodynamically stabilize the perovskite structure, which agrees with the previous experimental reports[10,41].

In principle, a lower $\triangle E_{\delta-\alpha}$ contributes to the higher purity of FA-based perovskite crystallization, leading to a lower percentage of the undesired yellow δ-phase. Additionally, $E_b$ dictates the phase transition rate during the device operation. These parameters, including $\triangle E_{\delta-\alpha}$ and $E_b$, are crucial indicators in describing the thermodynamics and kinetics of the phase transition in FAPbI₃ perovskites, relative to the initial performance and the stability of the related devices, respectively. The results, as shown in Fig. S2, highlight the importance of A-site cations in altering the structural transition thermodynamics of FAPbI₃.

In terms of kinetics, we noted that the magnitude of the $E_b$ range (Fig. 1d and Supplementary Fig. 2e, and refs. 9,10,15,39) is comparable to, and even slightly higher than, the diffusion barrier of the FA vacancy (~0.61 eV), which has been reported to be rather immobile in FAPbI₃ at room temperature[42]. This implies that even at temperatures slightly above room temperature, the α−δ reaction rate would be constrained by the intrinsic kinetic activation barrier. Consequently, there must be other factors responsible for the rapid phase transition of the FAPbI₃ perovskites typically observed in experiments.

### Defect-induced phase instability of α-FAPbI₃

Lattice imperfections are prevalent in hybrid perovskites due to their propensity for defect formation, especially the ones fabricated using solution-based processes[17]. In other words, the actual phase transitions certainly happen in the presence of defects. It is thus necessary to

evaluate how defects impact both the thermodynamics and kinetics of the phase transition. Among the various intrinsic point defects in FAPbI₃, we focus on those that are common and abundant in the lattice[43,44]. These typical defects are namely the I vacancy ($V_I^+$), I interstitial ($I_i^-$), Pb vacancy ($V_{Pb}^{2-}$), FA vacancy ($V_{FA}^-$), and the FA interstitial ($FA_i^+$), respectively[43,44].

Figure 2a, b shows the calculated phase transition barriers ($E_b$) and thermodynamics ($\triangle E_{\delta-\alpha}$) of the pristine FAPbI₃ and those containing native low-energy defects. It is found that the presence of $V_I^+$, $I_i^-$, and $V_{Pb}^{2-}$ reduces $E_b$, thereby accelerating the kinetics of the α-δ phase transition of FAPbI₃. However, these defects do not significantly impact $\triangle E_{\delta-\alpha}$. In contrast, FA-related defects, namely $V_{FA}^-$ and $FA_i^+$, play an insignificant role in affecting the kinetics of the overall structural transition as the calculated $E_b$ are close to that of the pristine case. Nevertheless, $V_{FA}^-$ and $FA_i^+$ can impact the thermodynamics of the phase transition by reducing and increasing $\triangle E_{\delta-\alpha}$, respectively.

Notably, $V_I^+$, $I_i^-$, and $V_{Pb}^{2-}$ defects share a common characteristic of being inherent to the inorganic Pb–I skeleton. In other words, the presence of these defects induces local symmetry breaking and introduces instability within the fundamental building block of the inorganic PbI₆ octahedra, which plays a dominant role in determining the phase transition barrier. In contrast, FA molecules undergo relatively simple migration and rotation and do not occur chemical bond stretching, breaking, and forming, which were found to overcome the rather low activation energy barrier during the structural transition[15,45,46].

To quantitively evaluate the effect of intrinsic defects on the lattice instability of FAPbI₃ under realistic conditions, we performed AIMD simulations on pristine α-FAPbI₃ and systems containing typical intrinsic defects at room temperature over a 30-ps period with a time step of 1 fs. While the simulations show no bond breaking and structure reconstruction for both defect-free and defective systems, the systems containing $V_I^+$, $I_i^-$, and $V_{Pb}^{2-}$ display a noticeable increase in the mean square displacements (MSD) of the inorganic Pb–I skeleton (Supplementary Fig. 3a). The enhanced MSD compared with the pristine system implies an enhanced vibrational motion of PbI₆

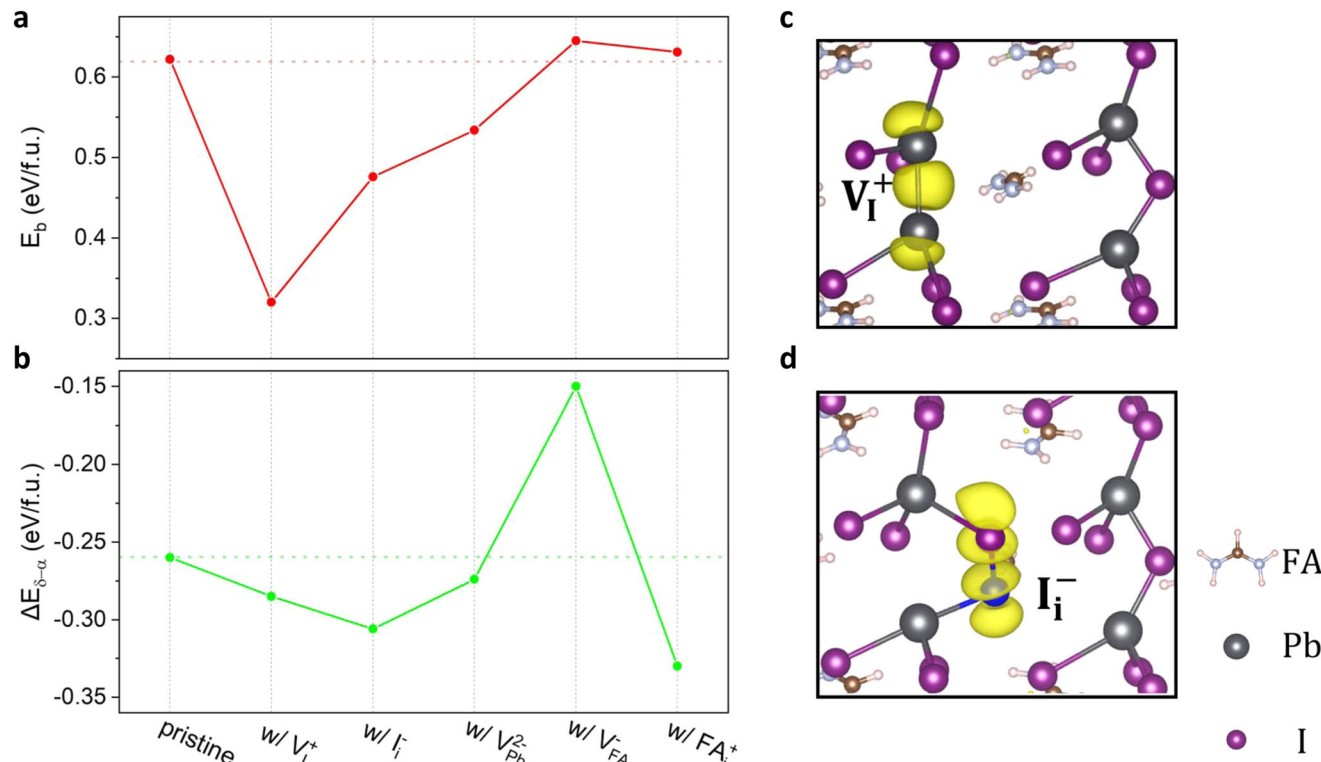

**Fig. 2 | Phase transition barriers and thermodynamics of FAPbI$_3$ with intrinsic defects. a** The phase transition barriers ($E_b$) and **b** total energy differences between α- and δ-phase ($\triangle E_{\delta-\alpha}$) of pristine FAPbI$_3$ and systems with the presence of the low-energy intrinsic defects. The charge density plots for the transition states of α-δ phase transition of FAPbI$_3$ with **c** V$_I^+$ and **d** I$_i^-$, where the yellow isosurfaces stand for the partial charge density for the states of the Pb dimer and I dimer. The isosurfaces were taken at 0.01 $e/\text{Å}^3$.

octahedron in the presence of V$_I^+$, I$_i^-$, and V$_{Pb}^{2-}$. As reported by Ghosh et al.[47], the unlocking of the PbI$_6$ octahedral vibration can hasten lattice breakage and tilting of the perovskites, thereby significantly reducing the structural dynamic stability.

Moreover, we noted that the systems containing V$_I^+$ and I$_i^-$ exhibit much lower $E_b$ (0.320 eV/f.u. for V$_I^+$ and 0.476 eV/f.u. for I$_i^-$) as compared to V$_{Pb}^{2-}$ (0.534 eV/f.u.). This can be attributed to the strong covalency at the transition states induced by V$_I^+$ and I$_i^-$ during the phase transition of FAPbI$_3$. In the case of V$_I^+$ (Fig. 2c), the two uncoordinated Pb cations near the vacancy move closer to form a Pb dimer with a strong covalent bond at the transition state. Similarly, I$_i^-$ induces a strong covalency between I anions that is characterized by the formation of an I dimer at the transition state (Fig. 2d). Such a covalent interaction would contribute to the stabilization of the configuration, resulting in lower energy for the transition state and consequently a reduced $E_b$. The defect-induced covalency has also been reported for iodine defects in hybrid perovskites like MAPbI$_3$[48] and inorganic perovskites like CsPbI$_3$[49] due to the soft lattice nature of halide perovskites. In comparison, no covalent bond forming was observed during the phase transition in the case of V$_{Pb}^{2-}$. Therefore, the combined effects of reduced lattice dynamic stability and defect-induced covalent interaction at the transition state are responsible for the substantially lower $E_b$ of FAPbI$_3$ perovskites with V$_I^+$ and I$_i^-$.

For V$_{FA}^-$ and FA$_i^+$, the MSDs of the inorganic Pb−I lattice are similar to that of pristine FAPbI$_3$ (Supplementary Fig. 3a), confirming that the FA-related defects have a negligible impact on the structural dynamic stability of the perovskite. This agrees with the small changes in $E_b$ obtained from the VCNEB calculations. On the other hand, as discussed above, the large size of FA cations is the key thermodynamic origin for the phase transition of FA-based perovskites, as also reflected by Goldschmidt's tolerance factor ($\tau = 1.02$) for FAPbI$_3$[50], which slightly exceeds the upper limit of 1 required for forming a stable perovskite structure. The presence of V$_{FA}^-$ increases the spatial cavity in α-FAPbI$_3$ and partially relieves the lattice strain induced by other large FA cations, thereby reducing the thermodynamic driving force for the conversion of the photoactive α-phase to the photoinactive δ-phase. Conversely, FA$_i^+$ aggravates the strained lattice, resulting in lower thermodynamic stability of the α-phase with respect to the δ-phase.

Phase transition typically proceeds through the growth and propagation of the targeted phase nuclei. Given that V$_I^+$ plays the most significant role in the local kinetic stability of FAPbI$_3$ with $E_b$ suffering the sharpest decline by -0.3 eV/f.u., we expected that the defects effectively facilitate the δ-phase propagation, responsible for a faster overall phase transition rate in FAPbI$_3$. To gain dynamic insights into the mechanism of phase transition aided by V$_I^+$ and verify our DFT results, we proceeded to construct a configuration with planar interfaces between the α-FAPbI$_3$ (111) and δ-FAPbI$_3$ (100) phases (Supplementary Fig. 4b), and perform 2-ns NpT Machine Learning Molecular Dynamics (MLMD) simulation at 300 K based on the training data set from on-the-fly hybrid AIMD. Energy calculations revealed that V$_I^+$ energetically prefers to reside at the interface than in the bulk of α-FAPbI$_3$ and δ-FAPbI$_3$ by 4.38 and 2.14 eV, respectively. Combing with the low diffusion barrier as previously reported[51], V$_I^+$ defects are expected to diffuse and aggregate around the interface between the two phases of FAPbI$_3$.

As shown in Fig. 3a, b, while no clear phase propagation was observed in the pristine system during the 2-ns MLFF simulation, a sign of notable transformation was evident in the V$_I^+$-defective system. Specifically, a PbI$_6$ octahedron at the interfacial plane around V$_I^+$ transformed to the face-sharing architecture of δ-FAPbI$_3$. This is further reflected by the analysis of MSD, as shown in Fig. 3c. The Pb−I inorganic skeleton on the interfacial plane exhibited a significantly

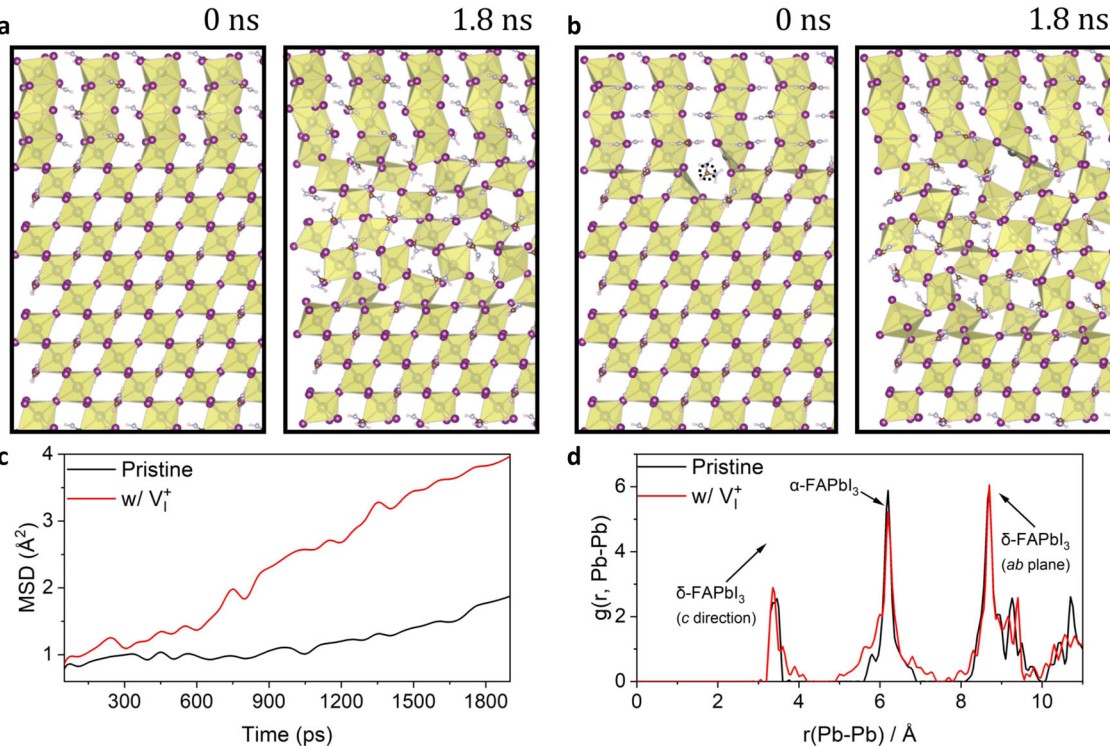

**Fig. 3 | MLMD simulation of FAPbI₃ α-to-δ phase propagation.** Snapshots captured at 0 and 1.8 ns from the Machine Learning Molecular Dynamics (MLMD) simulation of α-to-δ phase propagation in the **a** pristine and **b** $V_I^+$-defective planar interface models between α-FAPbI₃ (111) and δ-FAPbI₃ (100) at 300 K. **c** Mean square displacement (MSD) of the Pb–I inorganic skeleton of the interfacial plane (the region surrounded by the red dashed line in Supplementary Fig. 4b) and **d** the radial distribution function, $g(r, Pb - Pb)$, for Pb-Pb during the 2-ns MLMD simulations in the pristine and $V_I^+$-defective systems at 300 K.

larger MSD compared to that of the pristine systems throughout the simulation time. For a typical diffusive transformation associated with the α-δ phase transition in FAPbI₃[52], a larger MSD of the Pb−I inorganic skeleton in the $V_I^+$ system indicates a more diffusive host ion property around the two-phase interface. The increased diffusivity significantly promotes the local atomic rearrangement and reconstruction of the interfacial plane, thereby promoting phase propagation.

Moreover, to quantify these visual observations, we compared the radial distribution function (RDF) for Pb-Pb interaction in both the pristine and $V_I^+$-defective systems (Fig. 3d). For the ground-state atomic structures of α-FAPbI₃ and δ-FAPbI₃ (Supplementary Fig. 1), the second sharp peak corresponds to that in α-FAPbI₃, and the first and third sharp peaks stand for Pb-Pb along the c-direction and the ab-plane in δ-FAPbI₃, respectively. As can be seen, compared with the pristine system, the second peak in the $V_I^+$ system is broadened toward the first and third ones, accompanied by a slight reduction in coordination number. This implies a stronger tendency of the α-to-δ phase transition of FAPbI₃ in the $V_I^+$ system. Thus, a higher density of $V_I^+$ defects can result in an increased propagation rate of the undesired α-to-δ phase transition in FAPbI₃. The MLMD simulation provides a dynamic understanding of the DFT results.

Experiments reported that an excess of hole carriers can accelerate the phase transition in FAPbI₃[12,13], although the microscopic origin remains unclear. Our formation energy calculations show that a hole-rich environment enhances the density of detrimental $V_I^+$ defects (Supplementary Fig. 5). This suggests that the observed hole-rich-induced faster phase transition may be attributed to the increased density of $V_I^+$ defects in FAPbI₃[12,13]. While iodine-rich growth conditions can inhibit the formation of the most detrimental $V_I^+$ in FAPbI₃, they may also enhance the density of $I_i^-$, which can lead to a substantially reduced kinetic barrier of the phase transition. Hence, I-moderate growth conditions are preferable to balance the formation of these

two types of defects and to optimize the stability of FAPbI₃-based devices. Furthermore, the incorporation of certain additives, such as Lewis bases[53], to coordinate with the uncoordinated Pb of $V_I^+$ would also suppress the defect-accelerated phase propagation behavior. Moreover, the FA content should also be well controlled in the precursors, since the FA-rich growth conditions would promote the formation of FA interstitials in the perovskite lattice, which thermodynamically destabilizes the α-phase of FAPbI₃.

## Stabilization of α-FAPbI₃ by composition engineering

Doping engineering is a feasible strategy for improving the inherent phase stability of the FAPbI₃ active layers. To establish a rational and general design guideline for doping engineering in perovskites, we systematically explored the doping effects on stabilizing the phase of FAPbI₃ perovskites. We considered various isovalent substitution dopants on all the sites, namely A-site ($MA^+$, $Cs^+$, $Fr^+$, $Rb^+$, $K^+$, and $NH_4^+$), B-site (isovalent $Sn^{2+}$, $Ge^{2+}$, $Ba^{2+}$, $Sr^{2+}$, $Ca^{2+}$, $Cd^{2+}$, and $Zn^{2+}$, as well as Ln ions of $La^{3+}$, $Ce^{3+}$, $Nd^{3+}$, $Sm^{3+}$, $Eu^{3+}$, and $Yb^{3+}$) and X-site ($Br^-$, $Cl^-$, and pseudo-halides of $SCN^-$ and $CN^-$). These substitution dopants have been widely used in halide perovskites to tailor the optoelectronic and mechanical properties of the perovskite materials[4,6,7,22–24,54–56]. Additionally, we also considered unintentionally incorporated impurity interstitials, such as $H_2O$, $O_2$, $H_2$, and their disassociated products $OH^-$, $H^+$, $O^{2-}$. They are often found in the interstitial spaces of perovskite lattices[43,57–59].

Notably, unlike other isovalent candidates, Ln ions are often stable in the (+3) oxidation state, which can lead to charged defect formation when substituting $Pb^{2+}$. The formation energy calculations suggest the positively charged state of these substitutions is stable in FAPbI₃, namely $La_{Pb}^+$, $Ce_{Pb}^+$, $Nd_{Pb}^+$, $Sm_{Pb}^+$, $Eu_{Pb}^+$, and $Yb_{Pb}^+$ (Supplementary Fig. 6a). The substantially low formation energies, even in the negative region, over the whole range of the Fermi level within the bandgap,

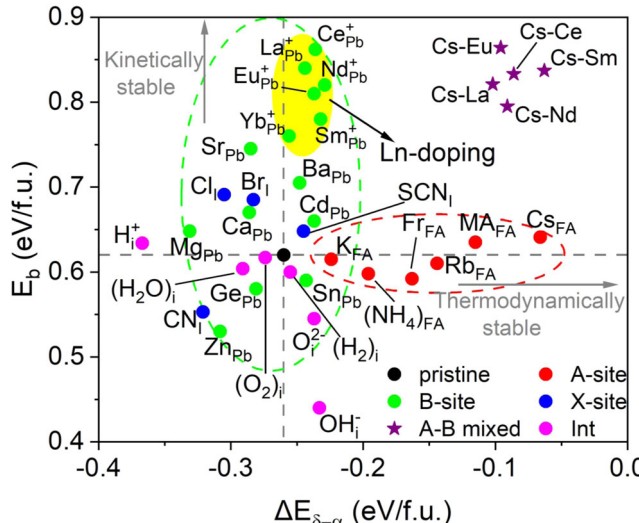

**Fig. 4 | Roles of doping and impurities in phase transition kinetics and thermodynamics of FAPbI₃.** Kinetic α-δ phase transition barrier $E_b$ and thermodynamic phase energy difference $\Delta E_{\delta-\alpha}$ of pristine FAPbI₃ and those with dopants and impurity interstitials. B-site cation engineering predominantly impacts the kinetics of the FAPbI₃ phase transition that dedicates the device longevity, as depicted by the green dashed circle (ellipse), whereas A-site doping more effectively alters the phase transition thermodynamics that can impact the perovskite crystallization, as depicted by the red dashed circle (ellipse). Moreover, the yellow region highlights various lanthanide doping and the purple stars stand for A-B mixed doping. The evolution of the bandgap and potential energy as a function of the reaction coordinate of the α-δ phase transition in the representative Cs−, Br−, Cl−, La−, and Ce-doped systems are shown in Supplementary Fig. 7.

further verify the doping feasibility of Ln cations in the FAPbI₃ lattice. Such a high compatibility can be partially attributed to their similar ionic radius to Pb. Furthermore, the calculated transition energy levels (Supplementary Fig. 6b) indicate that the Ln dopants have shallow defect states except for Yb, and thus are electronically benign for FAPbI₃.

Figure 4 summarizes the phase transition barriers $E_b$ and the phase transition thermodynamics $\Delta E_{\delta-\alpha}$ of pristine and doped (or defective) FAPbI₃. These intentional dopants and unintentional impurities exhibit varying effects on the phase transition of FAPbI₃. While atmospheric moisture-induced phase instability of FAPbI₃ has been experimentally reported, our results show that the water molecule itself has a negligible effect on the phase transition of FAPbI₃. Instead, the decomposition products of water, namely, OH⁻ and H⁺, greatly hasten the kinetics and thermodynamics of the phase transition, respectively (Fig. 4). The binding energy of H₂O with respect to isolated H⁺ and OH⁻ (H₂O → H⁺ + OH⁻) in FAPbI₃ was calculated to be −0.083 eV, implying that water disassociation in FAPbI₃ is energetically favorable. Light irradiation would facilitate this process and further aggravate the perovskite phase instability. Additionally, H₂ can be an effective hydrogen source for the incorporation of H⁺ in FAPbI₃[43], though it plays an insignificant role by itself. Moreover, O²⁻, which can be a disassociation product of atmospheric O₂ and various oxide electron-transporting materials, would also accelerate the FAPbI₃ phase conversion kinetics by reducing $E_b$. Therefore, in addition to humidity control, minimizing the concentration of oxygen and hydrogen ions in the lattice is important to stabilize the photoactive α-FAPbI₃.

Interestingly, A-site doping with smaller-sized cations can dramatically suppress the thermodynamic driving force for the phase transition while almost preserving the pristine transition barrier. Indeed, previous experiments have reported a higher initial

proportion of perovskite phase in the FAPbI₃ active layers when FA is doped with smaller-sized cations[60], due to the increased formation energy of δ-FAPbI₃. Such a mechanism is similar to the case of the FA vacancy as discussed earlier. Our results show that Cs, with a moderate ionic radius (1.74 Å), exhibits a higher effectiveness in increasing $E_b$, compared with that of MA (2.16 Å) and Fr (1.94 Å) of relatively larger ionic radius as well as the smaller NH₄ (1.43 Å), K (1.51 Å) and Rb (1.61 Å) ions. Hence, to crystallize a purer FA perovskite and thus obtain a higher initial performance of the devices, Cs would be a superior candidate for A-site cation engineering.

Among the X-site dopants, Br_I and Cl_I show a more significant improvement in the FAPbI₃ perovskite phase stability in kinetics, yet slightly increasing the thermodynamic energy difference between the two phases. This can be attributed to the improved stability of the sublattice as the result of the stronger Pb-Br and Pb−Cl bond than the Pb−I bond (because of the higher electronegativity of Cl and Br than I). In this case, more energy is required to convert the 3D octahedral structure to the 1D chain during the phase transition by stretching and breaking the stronger Pb−Br (Pb−Cl) bond. Experimentally, doping the FAPbI₃ active layers with MABr, CsBr, MACl, and CsCl additives has been a popular and efficient approach for stabilizing the α-phase FAPbI₃[6,11,22,23,25−28], though the underlying mechanisms and trends remain poorly understood[7,50]. Based on the findings discussed above, it is evident that Br and Cl increase the activation energy for the phase transition of FAPbI₃, while MA and Cs improve its thermodynamic stability. Both effects synergistically stabilize α-phase FAPbI₃. Moreover, our results suggest that pseudo-halide SCN⁻ would be also a promising X-site dopant for enhancing the perovskite phase stability of FAPbI₃ in terms of both thermodynamics and kinetics.

For B-site cation engineering, the dopants have a positive impact on the thermodynamics of the FAPbI₃ phase transition with the exception of Mg, Ca, Ge, and Zn, due to their much smaller ionic radii (0.72−1 Å) than that of Pb (1.19 Å), resulting in a greater deviation of tolerance factor from the ideal range of the perovskite. As a result, their inclusion can disrupt the stability of the perovskite structure. In agreement with our predictions, experimental reports have indicated that Ca doping promotes the formation of a non-perovskite δ-FAPbI₃ during the initial crystallization at the dopant concentrations of 5%[56]. In contrast, most B-site dopants exhibit higher effectiveness on the kinetics of the phase transition, leading to a larger variation in $E_b$, as indicated by the green dashed circle in Fig. 4.

Significantly, the Ln elements, which have been experimentally reported to be of benefit for the structural stability of halide perovskites[61], indeed represent ideal dopants of B-site doping for kinetically stabilizing α-FAPbI₃. As shown in the yellow region in Fig. 4, the incorporation of Ln dopants leads to a noticeable increase in $E_b$ with respect to the pristine FAPbI₃. The enhancement of $E_b$ upon Ln doping is of the order of pristine→Yb→Sm→Eu→Nd→La→Ce-doped systems. The enhanced transition barriers indicate improved lattice dynamic stability of α-FAPbI₃ upon Ln doping, which is also reflected by the reduced MSD of the inorganic skeleton in AIMD at 300 K for the Ln_Pb-incorporated systems (Supplementary Fig. 3b). This is because of the stronger ionic interactions between Ln ions and the nearby I ions against large octahedral tilt and rotation during the phase transition compared with that of Pb, due to the much lower electronegativities of Ln (1.1−1.2) than that of Pb (2.3). Taking Ce as an example, a more pronounced charge redistribution occurs in CeI₆ octahedra than in PbI₆ (Fig. 5a), confirming the stronger ionic bonding of Ce−I compared to the corresponding Pb−I in pristine FAPbI₃.

From a thermodynamic perspective, Ln-cation doping would also effectively reduce the driving force for the transition to the photoinactive δ-phase of FAPbI₃ as compared with other B- and X-site dopants, particularly for Nd and Sm-doped systems, even though the

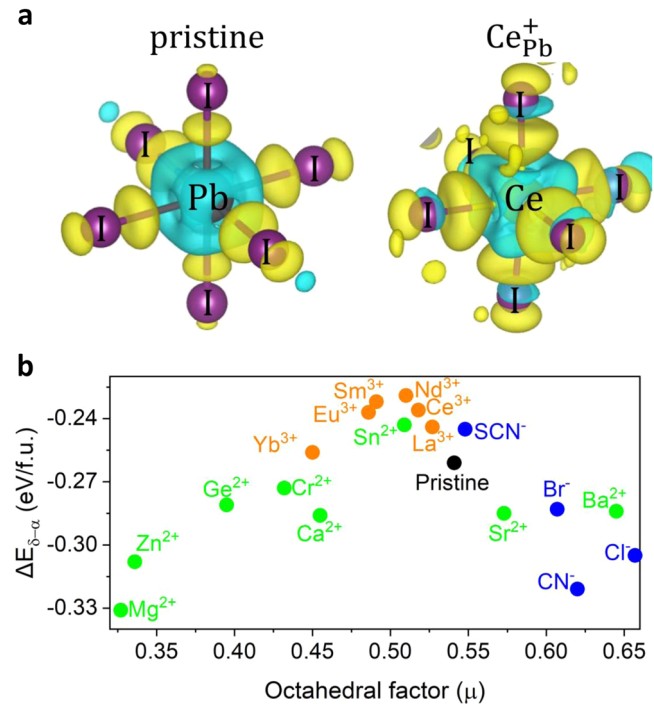

**Fig. 5 | Charge density differences and phase transition thermodynamics of the doped FAPbI₃. a** Calculated isosurfaces of the charge density differences for PbI₆ and CeI₆ octahedra, where electrons are transferred from the yellow to the blue regions. The isosurfaces were taken at 0.01 $|e|/\text{Å}^3$. **b** The phase transition thermodynamics $\triangle E_{\delta-\alpha}$ of the B- and X-site doped FAPbI₃ as a function of the octahedral factor ($\mu$), where the blue points stand for X-site isovalent dopants, green for B-site isovalent dopants, and orange highlights Ln dopants.

effectiveness is relatively smaller than that of the most A-site dopants. Regarding B- and X-site doping engineering, Fig. 5b illustrates a correlation between the thermodynamic driving force variation and the octahedral factor ($\mu$) of Pauling's rule, which is an important factor determining the coordination number of the cation and anion in stable crystal structures and is defined as the ratio of the radii of B cations and X anions in perovskites, namely $\mu = \frac{r_B}{r_X}$. In general, to stably construct the perovskite compounds of the six-fold coordinated octahedra, it requires $0.414 < \mu < 0.732$. Based on the present volcano-type relationship, we deduced that the optimal value $\mu$ (apex) for the stable FAPbI₃ perovskites phase would be around 0.5. Sn and Ln dopants, especially Nd, Eu, and Sm, are close to this optimal value of $\mu$. Indeed, Sn halide perovskites typically exhibit higher phase stability in thermodynamics[62]. However, the facile oxidation of $Sn^{2+}$ poses a serious impediment to the further development of Sn-based and Sn-doped perovskite optoelectronics operating in ambient air. B-site doping with Ln ion represents a potential breakthrough for further stability improvement of FAPbI₃ perovskites and devices.

In addition to defect and impurity controls, the current results underline the effectiveness of proper doping engineering in stabilizing α-FAPbI₃. Particularly, B-site doping primarily impacts the kinetics of the FAPbI₃ phase transition, while A-site cation engineering is more effective in thermodynamics in controlling the phase stability of FAPbI₃. We thus suggest a highly stable FAPbI₃ perovskite active layer and the related devices by A-B mixed composition engineering from both thermodynamic and kinetic perspectives. This is confirmed by calculations demonstrating the synergic stabilizing effects of the exemplary Cs-La, Cs-Ce, Cs–Nd, Cs–Eu, and Cs–Sm-doped systems (see the purple stars in Fig. 4). Promisingly, based on Shockley–Queisser (SQ) limit[63], these doped perovskites still maintain the superior electronic bandgaps, which has direct implications for the

observed efficiency in the conversion of solar light (Supplementary Fig. 8). Moreover, the findings also explain why state-of-the-art perovskite solar cells are normally based on mixed systems with the majority of FAPbI₃ perovskite[1,3–8], which synergistically stabilize FAPbI₃ perovskite and is essential for achieving long-term device operational stability.

We noted that the majority of the proposed A-B mixed-doped FAPbI₃ perovskites denoted by the purple stars in Fig. 4 have not yet been reported. To validate our theoretical findings, we pursued experimental studies on Cs-Eu mixed doping in FAPbI₃ as a representative example, which is predicted to significantly stabilize the α- FAPbI₃ - both kinetically and thermodynamically. To this end, we first grew single crystals of Cs-doped and Cs-Eu mixed-doped FAPbI₃ perovskites (Supplementary Fig. 9), and the related synthetic details are shown in the Experimental Method section. Energy-dispersive X-ray spectroscopy (EDS) measurement results on the Cs-Eu mixed-doped FAPbI₃ single crystal confirm the uniform distribution of the Eu element in the perovskite (Supplementary Fig. 10).

## Stability and optoelectronic properties of Cs-Eu mixed-doped perovskite

To assess the phase stability in the Cs−Eu mixed-doped perovskite single crystal, we then exposed the samples to air for 30 days. Initially, both as-grown samples (namely, Cs-doped and Cs−Eu mixed-doped FAPbI₃) exhibited black perovskite phases (α-phase) with high purity, as confirmed by the relevant X-ray diffraction (XRD) patterns (see Fig. 6a, b). After the 30-day exposure, the Cs-doped sample visibly degraded to the yellow δ-FAPbI₃ phase (Supplementary Fig. 10). In contrast, the Cs-Eu mixed-cation-doped perovskite largely maintained the black α-FAPbI₃ phase without significant changes in color. This trend was also evident in the XRD spectra of the samples measured after 30-day and 45-day exposures. Figure 6c−e shows the UV-Vis-NIR absorption spectra of the freshly prepared samples with different concentrations and those after the 30-day exposure. The optical bandgap derived from the Tauc plots exhibited a value of 1.50–1.52 eV for Cs−Eu mixed-doped FAPbI₃ (Fig. 6d), in agreement with the value of 1.50 eV predicted by our HSE-SOC calculations. Indeed, compared with the Cs-doped FAPbI₃, we note that the bandgap and absorption profile of the Cs−Eu mixed-doped sample showed negligible change after 30-day exposure. Specifically, the average absorbances at 790 nm after 30-day exposure were dropped by 28.5% and 5.2% for Cs-doped and Cs-Eu mixed-doped (0.5% Eu doping) FAPbI₃ perovskite single crystals, respectively (see Fig. 6e). Moreover, we investigated the phase stability improvement of the Cs−Eu mixed samples as a function of Eu dopant concentration. With increasing the concentration of Eu doping, the degradation of single crystal perovskite optical properties was effectively suppressed (Fig. 6e). The optimal concentration of Eu doping was tested to be 0.5%.

We further investigated the carrier mobility and the trap density, which are the key semiconducting parameters for optoelectronic applications, by using the space-charge-limited current (SCLC) method[64] based on the hole-only device (see Supplementary Note 1). Previous experimental studies[65] have shown that α-δ phase transition in FAPbI₃ leads to a significant degradation in carrier transport property. Indeed, our DFT calculations of the effective masses of electrons and holes in the yellow δ-phase ($m_e^* = 0.945m_0$; $m_h^* = -1.137m_0$), are around 4.3 and 5.7 times higher than those in black α-phases of FAPbI₃ ($m_e^* = 0.218m_0$; $m_h^* = -0.199m_0$), respectively, also as explicitly evidenced by the substantially flatter band edges in δ-FAPbI₃ (Supplementary Fig. 1c, e). Figure 6f−i shows the current-voltage (I–V) characteristics for the Cs single-doped and Cs−Eu mixed-doped FAPbI₃ single crystals in both fresh statues and after 30 days. The I−V curves can be divided into three regions: the first, second, and third regions stand for the ohmic ($n = 1$), trap-filling ($n > 3$), and child ($n = 2$) regions,

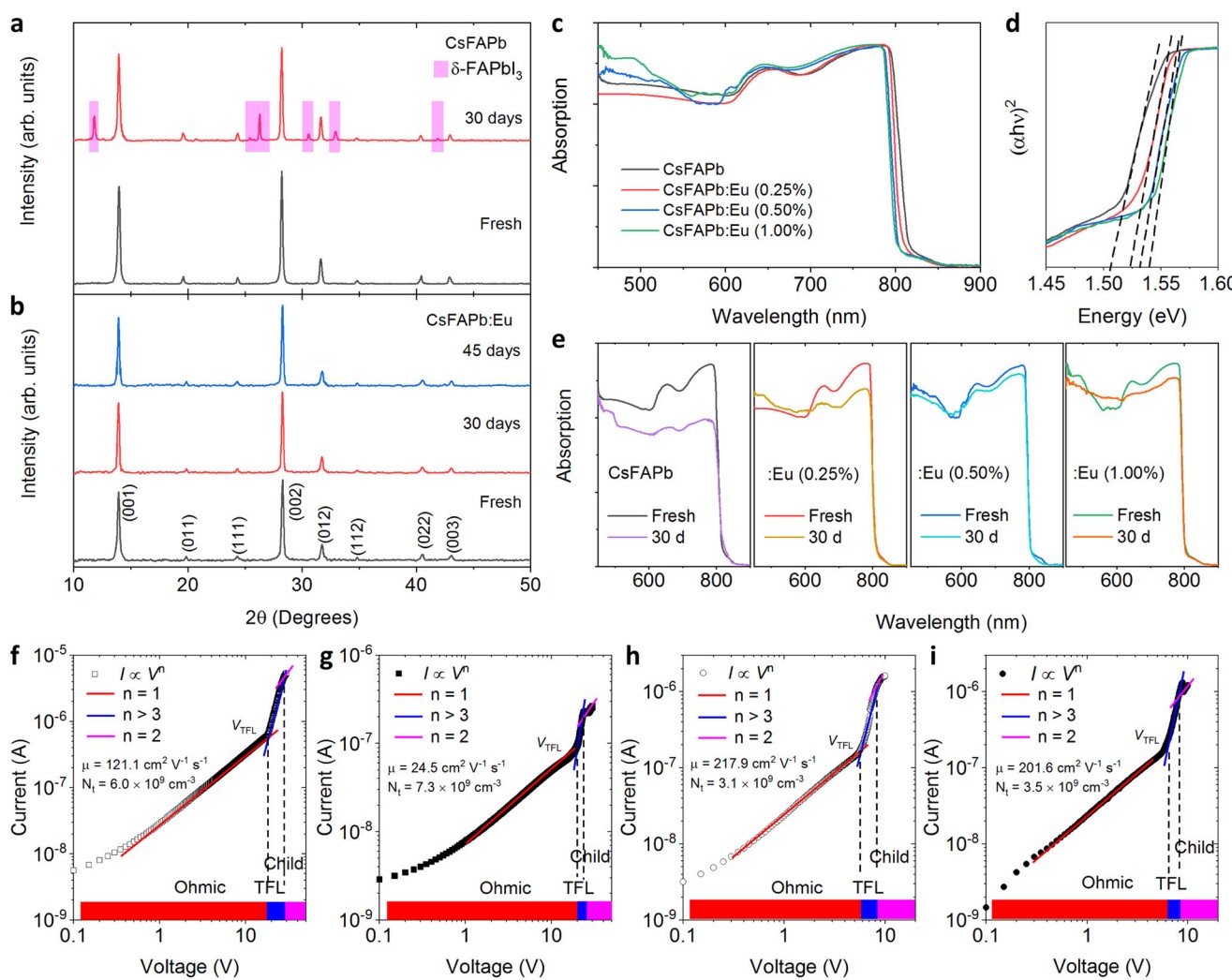

**Fig. 6 | XRD, absorption spectra, and I–V characteristics of Cs-doped and Cs–Eu-doped FAPbI₃ single crystals.** X-ray diffraction patterns of the **a** Cs-doped and **b** Cs-Eu mixed-cation-doped FAPbI₃ perovskite single crystals, before and after 30-day and 45-day air exposures. **c** UV–Vis–NIR absorption spectra of a series of freshly prepared perovskite single crystals. **d** The optical bandgap derived from the Tauc plots (based on the data in **c**). α is absorption coefficient, h is Planck constant, and ν is frequency of incident light. **e** UV–Vis–NIR absorption spectra of the Cs-doped and Cs-Eu mixed-cation-doped samples with the Eu doping concentrations of 0.25%, 0.50%, and 1.00% before and after 30-day exposure. Current–voltage (I-V) curves and relevant typical SCLC analyses of **f**, **g** Cs single-doped and **h**, **i** Cs–Eu mixed-doped FAPbI₃ perovskite single crystals, before and after a 30-day air exposure, respectively. The regions are marked for Ohmic (n = 1), trap-filled limit (n > 3), and Child's regime (n = 2). $V_{TFL}$ is the trap-filled limit voltage. We calculated the carrier mobility (μ) and trap densities ($N_t$) by fitting the I-V data.

respectively, where the trap density ($N_t$) and carrier mobility (μ) were calculated from the second and third regions (see the details in Supplementary Information). It is noted that, as compared with the Cs-doped sample ($n_t \sim 6.0 \times 10^9$ cm$^{-3}$; $\mu \sim 121.0$ cm$^2$ V$^{-1}$ s$^{-1}$), a lower trap density ($n_t \sim 3.1 \times 10^9$ cm$^{-3}$) and a higher hole mobility ($\mu \sim 217.9$ cm$^2$ V$^{-1}$ s$^{-1}$) were obtained for the freshly prepared Cs-Eu mixed-doped perovskite single crystal, confirming the promoting effect of Cs–Eu mixed doping on carrier dynamics of FAPbI₃ perovskites. After 30-day exposure, we observed a significant decline of hole mobility in Cs-doped sample ($\triangle\mu \sim 96.5$ cm$^2$ V$^{-1}$ s$^{-1}$), as compared to that of Cs–Eu mixed-doped sample ($\triangle\mu \sim 16.3$ cm$^2$ V$^{-1}$ s$^{-1}$), highlighting the effectiveness of A-B mixed composition engineering in stabilizing FAPbI₃ perovskite.

## Discussion

In summary, based on extensive first-principles atomistic calculations, we have elucidated the intricate mechanism underlying phase degradation from the photovoltaically active α phase to the inactive δ-phase in FAPbI₃. Our investigation not only sheds light

on the critical role played by defects and impurities but also delves into potential mitigation strategies. Inherent iodine defects, namely iodine vacancies and interstitials, dramatically lower the activation barrier of the α-δ lattice transition in FAPbI₃, due to the compromised lattice dynamic stability and the strong covalency induced by defects at transition states. The detrimental roles of the decomposition products of atmospheric moisture and oxygen in degrading the FAPbI₃ perovskite phase were also identified, rationalizing the faster phase transition of FAPbI₃ films under ambient air. Notably, we uncovered key compositional design principles: B-site cation engineering is a promising strategy to tailor the kinetics, whereas A-site doping predominantly impacts the thermodynamics of the phase transition of FAPbI₃, highlighting the synergetic perovskite phase stabilization role of A-B mixed-site doping. A significant correlation was discovered between the phase transition thermodynamics and the octahedral factor (μ) of Pauling's rule, underlining the optimal value of $\mu \sim 0.5$ for stable FA-based perovskites and suggesting co-Ln doping as a promising strategy for B-site cation engineering.

Subsequent experimental findings support the prediction that Cs–Eu mixed-doped FAPbI₃, a theoretically identified superior doping candidate, exhibits superior absorption and carrier transport properties and greatly improved phase stability as compared to the Cs-doped counterpart. Overall, our study provides insights into defect control and synergetic composition engineering for developing FAPbI₃-based perovskite solar cells and other optoelectronic devices with superior initial device performance and highly improved long-term stability.

## Methods

### First-principles calculations
The first-principles calculations were performed based on the density functional theory (DFT) with the projector-augmented wave method as implemented in the Vienna Ab initio Simulation Package (VASP)[66]. The DFT-D3 scheme of Grimme was adopted for the van der Waals correction[67]. The kinetic energy cutoff of the plane-wave basis was set at 400 eV and a Monkhorst–Pack sampling of $2 \times 2 \times 2$ k-points was used for the $3 \times 3 \times 2$ supercells with 216 atoms. For the exchange-correlation functional, electronic structures, defect formation energies and transition energy levels were obtained using Heyd–Scuseria–Ernzerhof hybrid functional (the mixing parameter of 0.46) with the spin–orbit coupling (HSE-SOC)[68], and the VCNEB calculation, AIMD and MLMD simulations were performed based on the generalized gradient approximation of Perdew, Burke, and Ernzerhof (PBE)[69]. All the atomic positions were fully relaxed until the residual forces were less than 0.01 eV/Å.

### Activation energy barriers
The activation energy barriers ($E_b$), defined as the energy difference between the α-phase FAPbI₃ and the configuration at the saddle point along the α–δ phase transition pathway, were calculated using the VCNEB[37], which is an extension of elastic band method (NEB) technique, as implemented in the Universal Structure Predictor: Evolutionary Xtallography (USPEX) code in combination with the VASP. Compared with the traditional NEB algorithm, the VCNEB technique allows both the atomic positions and the supercell dimensions to relax for each configuration at constant pressure, thereby having been widely used for the analysis of the solid–solid phase transition pathways and activation energy barriers in both the hybrid perovskite systems[15,39] and other materials[70,71]. Note that the simulated lowest-energy transition path is similar to that reported in ref. 15. The effect of intrinsic defects, extrinsic doping, and unintentional impurities on the phase transition thermodynamics and kinetics was evaluated based on the $3 \times 3 \times 2$ supercells with 216 atoms (defect concentration of 5.56%).

### Ab initio molecular dynamics
AIMD simulations lasted 30 ps with the time step of 1 fs in the canonical (NVT) ensemble. The temperature was controlled at 300 K by using the Nose–Hoover thermostat[72].

### Machine learning molecular dynamics
MLMD[73,74] was performed as implemented in VASP. All the datasets were trained based on the on-the-fly hybrid AIMD and MLMD calculations, in which first-principles ab initio calculations will be conducted when local molecular environments are significantly different from those already stored as the training data. In this study, Machine Learning Force Field (MLFF) was trained in an isobaric–isothermic (NpT) ensemble with a time step of 1 fs, using the $(3 \times 3 \times 2)$ 216-atom bulk supercells of α-FAPbI₃ and δ-FAPbI₃, 360-atom surface models of α-FAPbI₃ (111) and δ-FAPbI₃ (100) with a vacuum thickness of 15 Å, 312-atom (Fig. S4a) and 1248-atom (Fig. S4b) interface models between the α-FAPbI₃ (111) and

δ-FAPbI₃ (100) phases, and those with a $V_I^+$. This interface model has been reported to exhibit the lowest total energy[75], and the lattice mismatch was less than 1%. The force field was generated with a cutoff radius of 7 Å for the angular descriptor and a width of 0.5 Å of Gaussian functions for broadening the atomic distributions of the radial descriptor. The MLFF training involves 10-ps NpT simulations at 200 K, 300 K, and 400 K for each configuration. Based on the obtained datasets of MLFF, 2 ns NpT-MLMD simulations with a time step of 1 fs were run for the 1248-atom interface models of pristine FAPbI₃ and that containing a $V_I^+$.

### Defect formation energy
The defect formation energy was calculated by ref. 17:

$$\triangle H_f(X_i^q) = E(X_i^q) - E(\text{host}) - \sum n_i(\mu_i + \triangle\mu_i) + q(E_f + E(\text{VBM}) + \triangle V) + \triangle_{\text{corr}}^q \tag{1}$$

where $E(X_i^q)$ and $E(\text{host})$ are the total energies of the defective and defect-free supercells, respectively. $n_i$ stands for the number of defects added into the supercell. $\mu_i$ is the absolute value of the chemical potential of the defect atoms, and $\triangle\mu_i$ the relative value of the chemical potential, which is related to growth conditions. For the host chemical potentials, we chose that under the iodine-moderate conditions (see Supplementary Fig. 11), which are representative of the typical synthesis. The chemical potentials of Ln elements correspond to those in lanthanide iodide salts LnI₃. $E(\text{VBM})$ is the energy of the valance band maximum (VBM) of CsPbI₃ and $E_f$ represents the Fermi energy measured from the VBM. $\triangle V$ is the correction term for ensuring the alignment of the potential for the charged defect in supercells, and the finite-size correction term, $\triangle_{\text{corr}}^q$, was used for correcting the periodic images of the charged defects. Moreover, a convergence test was performed using a $3 \times 3 \times 3$ 324-atom supercell of FAPbI₃. The results show that, for $V_I^+$, the calculated formation energy difference between the 216-atom and 324-atom supercells is only 0.053 eV.

### Defect concentration
Based on the formation energy, the defect concentration at thermal equilibrium thus can be given by ref. 17: $N = N_0 e^{-\frac{\triangle H}{kT}}$, where $N_0$ stands for the number of available sites for defect formation in the CsPbI₃ lattice per volume, $\triangle H$ for the formation energy of defect, $k$ for the Boltzmann constant, and $T$ is temperature.

### Transition energy level
The transition energy level $\mathcal{E}(q/q\prime)$ was determined by the Fermi level position for which the formation energy of $X_i^q$ is equal to that of $X_i^{q\prime}$, thus, can be determined by ref. 17:

$$\mathcal{E}(q/q\prime) = [E(a,q) - E(a,q\prime) - (q - q\prime)(E(\text{VBM}) + \Delta V)]/(q - q\prime) \tag{2}$$

### Crystal growth
All the series of mixed-cation Pb-based perovskite single crystals were grown using the inverse temperature crystallization method. Specifically, a stoichiometric molar ratio of CsFA precursor solution (1.0 M) was prepared by dissolving 1.55 g FAI (9.0 mmol), 0.26 g CsI (1.0 mmol), 0.55 g PbBr₂ (1.5 mmol), and 3.85 g PbI₂ (8.5 mmol) compounds in 10 ml γ-butyralactone. After stirring at room temperature for 12 h, the mixed solution was filtered through a 0.22 μm polytetrafluoroethylene filter and transferred into glass vials. Thereafter, the glass vials with the filtered solution were placed on the hot plate heated to 80 °C and kept this temperature for 3 h. Then, the temperature was gradually increased from 80 °C to 120 °C at a slow rate of 2 °C/h. Small crystals would be formed in the crystallizing dish. For Cs/Eu-

doped mixed-cation perovskite single crystal, a small amount of $EuI_3$ was dissolved into the solution of $Cs_{0.1}FA_{0.9}Pb(I_{0.9}Br_{0.1})_3$ at a specific concentration. The amounts of B-site dopants (Eu: 0.25%, 0.50% and 1.00%) in the perovskite solution are calculated according to the molar ratio with Pb.

## Device fabrication

The hole-only device with vertical Au/Perovskite/Au structure was prepared for the SCLC method. An 80-nm gold (Au) anode was deposited on the top surface of the mixed-cation perovskite single crystal by thermal evaporation (AJA ATC-1800-E e-beam thermal evaporator). Then, a 105 nm Au cathode was deposited on the bottom surface of the single crystal samples. The obtained devices will be used for the SCLC measurements.

## Characterizations

The XRD patterns of the perovskite powder (by gridding the single crystals into powder) were measured using on PANalytical Pro Powder Diffractometer with a Cu Kα radiation source. The absorption spectrum measurement was taken in an Agilent Cary 5000 UV–vis–NIR spectrophotometer. EDS testing were carried out using Zeiss ULTRA plus under the operating voltage of 20 kV. The *I-V* characteristic curves of perovskite single crystals were measured using a two-terminal probe station (Everbeing Int'l Corp.) and 4200A-SCS Parameter Analyzer.

## Data availability

Data to support the findings and conclusions are included in the published article and its Supplementary Information and Source Data files. Source data are provided with this paper.

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

## Acknowledgements
We acknowledge the expert support provided by the Sydney Informatics Hub (SIH) team—a core research facility of the University of Sydney. This work was supported by computational resources provided by the Australian Government through Gadi under the National Computational Merit Allocation Scheme and was accessed through the SIH HPC Allocation Scheme [LE190100021]. We acknowledge partial financial support from the Australian Research Council [DP200100940] and [DE180100167] and The University of Sydney Physics Foundation.

## Author contributions
Y.L., X.C. and R.Z. conceived the idea. Yuhang Liang performed the theoretical simulations under the supervision of X.C., J.H. and R.Z. F.L. carried out the experimental verification. T.L., C.S., S.R., X.Y. contributed to the discussion and writing of the paper.

## Competing interests
The authors declare no competing interests.
