## [Peer Review File · Nature Communications]

REVIEWERS' COMMENTS

Reviewer #1 (Remarks to the Author):

In this work, the authors employed density functional theory (DFT) to study on the phase stability of FAPbI₃. Authors demonstrated that intrinsic defects can significantly expedite the structural transition kinetics and found that the detriment of moisture and oxygen on the phase stability of FAPbI₃. Finally, authors proposed the defect control and composition engineering to realize the stabilization of FAPbI₃. However, there are numerous literatures have demonstrated the effectiveness of A-site engineering on achieving phase stable FAPbI₃. Moreover, although authors claimed that B-site dopant can successfully manipulate the kinetics of the phase transition in FAPbI₃, there is no experiment results in this manuscript to support that. Therefore, this work is not convincing enough because it is just based on calculations without experimental results. I may reconsider the publication of this work after adding the related experimental work to support the calculation results.

1. There are many literatures [Kim et al., Joule 3,1-14; Lu et al., Science 370, 74 (2020); Nature 2021, 592, 381; Nature Energy 6, 419-428 (2021)] have done both theory and experiment work to study the role of A-X doping and I vacancies on the stability of FAPbI₃. Therefore, this work compared with above literatures does not exhibit enough novelty to publish on Nature communications.

2. Authors claimed that the uncoordinated Pb induced by I vacancies can form Pb dimer to reduce the Eb. Moreover, as mentioned by authors, the mitigated method of I vacancies by growth of perovskite with I rich condition will lead to enhance the density of I interstitials. Therefore, can we just use some additives, for example Lewis base, to coordinate with uncoordinated Pb and eventually enhance the Eb?

3. Interestingly, authors demonstrated that inclusion of Ca in perovskite results in disrupting the stability of the perovskite structure. However, the experiment results from literature [Nature Materials 21, 1396-1402 (2022)] have showed that the dopant of Ca contribute to enhancing the stability of perovskite films and photovoltaic performance of devices, which seems to be contradictory with the conclusion in this work. Can the author explain it?

4. Although the phase transition barriers and the phase transition thermodynamics ΔE of pristine and doped FAPbI₃ have been summarized at Figure 3. But the detailed evolution of the lattice parameters, volume, bandgap, and the potential energy as a function of reaction coordinate of the α - δ phase transition in Cs doped FAPbI₃, FAPbI₃ with BrI and ClI defect and some typical B-site doped FAPbI₃ still need to be provided.

5. The authors have shown the phase transition thermodynamics of the B- and X-site doped FAPbI₃ as a function of the octahedral factor. For me, the linear relation between octahedral factor and phase transition thermodynamics ΔE is just too good to be true.

6. The calculations in this work demonstrating the synergic stabilizing effects of the exemplary Cs-La and Cs-Ce doped systems. However, why not provide the experimental results to further support the effectiveness of A-B mixed dopant on phase stability of FAPbI₃. We believe that the author should provide the corresponding experimental results to determine the reliability of the calculation results, which also meets the requirements of the journal.

7. I think the doping concentration will also play a crucial role for the phase stability. But there is no detailed discussion for the concentration in this work.

Reviewer #2 (Remarks to the Author):

The article entitled "Toward stabilization of formamidinium lead iodide perovskites by defect control and composition engineering" addresses the challenge of phase instability in FAPbI₃-

based perovskite solar cells. The authors apply computational simulations to investigate the mechanism behind the undesired α - δ phase transition of FAPbI_3 . Iodide vacancies and interstitials are found to significantly impact the transition kinetics by inducing covalency during transition states. The study also highlights the detrimental effects of atmospheric moisture and oxygen on the perovskite phase. Furthermore, the authors propose A-B mixed doping could be potentially an effective strategy for stabilizing the desired α -phase.

The systematic study on different defects and dopings in this work demonstrates a thorough investigation of the factors influencing phase instability in FAPbI_3 -based perovskite solar cells. By elucidating the role of these defects, the authors provide important guidance for defect control strategies and potential avenues for improving the stability of FAPbI_3 perovskite materials. As such, I am supportive of publication and would suggest a few points that the authors could consider to discuss in the next revision.

The results in Fig. 3 suggest that B-site engineering significantly influences the kinetics of the phase transition, while A-site engineering primarily alters the thermodynamics. The study further demonstrates the synergistic stabilizing effects of Cs-La and Cs-Ce mixed doping, leading to a highly stable FAPbI_3 perovskite. However, it is recommended that the authors include information on the optoelectronic properties of these mixed-doped perovskites.

Considering the potential impact of B-site doping on the electronic properties, it is essential to evaluate the suitability of these materials for photovoltaic applications. The inclusion of optoelectronic property data would further enhance the significance and applicability of the study. Further to the first question, I wonder if the authors considered different A-B mixed doping configurations in their study. Specifically, did they explore multiple combinations of A-site and B-site dopants? Moreover, it would be interesting to know if E_b and $\Delta E_{\alpha-\delta}$ vary across these different configurations.

The authors further conducted AI-MD simulations to investigate the mechanism driving the α - δ phase transition of FAPbI_3 . However, considering the relatively short simulation time, the results obtained from these simulations may not provide conclusive evidence. Therefore, I would recommend moving this section to the supporting information.

Reviewer #3 (Remarks to the Author):

In the manuscript, the author presents extensive first-principles atomistic calculations of the phase transition of FAPbI_3 from α -phase to δ -phase. However, despite the author's thorough computation and comparison of activation energy barriers throughout the manuscript, the primary conclusions rely solely on these activation energy barrier calculations. This lack of cross-verification with multiple sets of computational results undermines the solidity of the author's conclusions. For example, on page 10 line 269-272, the author mentions, "Given that V_{I^+} defects play the most significant role in the local kinetic stability of FAPbI_3 with a sharp decline of ~ 0.3 eV/f.u., nucleation for the phase transition is likely to occur around the defect centers due to a much faster nucleation rate." I do not believe that such a conclusion, particularly regarding nucleation, can be drawn solely from an energy perspective. In reality, the nucleation process is a complex phenomenon influenced by multiple factors of both kinetics and thermodynamics. It would require additional verification through dynamic nucleation simulations, such as those conducted using AIMD (Ab Initio Molecular Dynamics) methodology or Classical MD. Performing additional phase transition simulations based on AIMD may offer more dynamic insights and information, potentially addressing the limitations of this manuscript, which heavily relies on DFT static calculations. Additionally, the MSD curve in Figure 2(c) does not reach a linear trend, showing unstable or even decreasing trends. This is likely due to restricted ion movement in confined lattice structure, rendering the MSD calculation data unsuitable for supporting the conclusions presented in page 9 256-258. A near flat MSD curve implies that the particle is essentially confined to lattice sites and undergoes localized vibrations. Based on the data presented in the manuscript so far, the support for the results is not sufficiently solid and comprehensive. Therefore, I recommend that this article is not suitable for publication in its current form.

RESPONSE TO REVIEWERS' COMMENTS

Below, please find our point-to-point response to all the comments. The changes and revisions in the updated manuscript are highlighted in red

Reviewer #1

Comment: In this work, the authors employed density functional theory (DFT) to study on the phase stability of FAPbI₃. Authors demonstrated that intrinsic defects can significantly expedite the structural transition kinetics and found that the detriment of moisture and oxygen on the phase stability of FAPbI₃. Finally, authors proposed the defect control and composition engineering to realize the stabilization of FAPbI₃. However, there are numerous literatures have demonstrated the effectiveness of A-site engineering on achieving phase stable FAPbI₃. Moreover, although authors claimed that B-site dopant can successfully manipulate the kinetics of the phase transition in FAPbI₃, there is no experiment results in this manuscript to support that. Therefore, this work is not convincing enough because it is just based on calculations without experimental results. I may reconsider the publication of this work after adding the related experimental work to support the calculation results.

Our reply: We thank the reviewer for careful reading and the constructive suggestions on our work, and for the possibility of reconsidering the publication of this work after adding the related experimental results. In this sense, we have added the related experimental data in the updated manuscript.

(1) There are many literatures [Kim et al., *Joule* 3,1–14; Lu et al., *Science* 370, 74 (2020); *Nature* 2021, 592, 381; *Nature Energy* 6, 419–428 (2021)] have done both theory and experiment work to study the role of A-X doping and I vacancies on the stability of FAPbI₃. Therefore, this work compared with above literatures does not exhibit enough novelty to publish on *Nature* communications.

Our reply: We thank the reviewer for sharing the important and relevant papers. Given the phase stability of FAPbI₃ perovskite is an outstanding challenge, many papers have done both the experimental and theoretical studies aiming to improve the phase stability of FAPbI₃ perovskite mainly through A- and X-site doping, including MAI doping (*Joule* 3, 2179–2192 (2019)), SCN⁻ doping (*Science* 370, (2020)), HCOO⁻ doping (*Nature* 592, 381–385 (2021)), and iPA_mHCl doping (*Nat. Energy* 6, 419–428 (2021)). We have cited these published papers, including *Nature* 592, 381–385 (2021) and *Science* 370, (2020) as ref. 2 and ref. 4 in the original manuscript. In the updated manuscript, we also cited *Joule* 3, 2179–2192 (2019) and *Nat. Energy* 6, 419–428 (2021) as ref. 27 and ref. 28, respectively.

It is worth noting that previous studies have mainly focused on A-X doping on specific doping systems, falling short of a comprehensive understanding of the general doping

principles for each atomic site (A-, B-, and X-site) in stabilizing FAPbI₃ perovskites. Therefore, in our manuscript, we have systematically studied the stabilization mechanisms across all atomic-site doping in FAPbI₃ (including both the widely employed and unexplored dopant species). We discovered the general doping design principles general for stabilizing α -FAPbI₃: A-site engineering predominantly influences the thermodynamics and B-site doping (in particular the doping of Lanthanide cation) is more effective in manipulating the kinetics of the phase transition in FAPbI₃. Thus, A-B mixed cation engineering emerges as a potent approach for synergistic FAPbI₃ perovskite stabilization. These insights, previously unreported, provide a key scientific foundation for future rational doping in stabilizing FAPbI₃ perovskites and related devices.

Importantly, as the reviewer suggested, we have performed the related experimental verification of B-site doping to bolster the reliability of the theoretical conclusions. Further details can be seen in the reply to question 6.

(2) Authors claimed that the uncoordinated Pb induced by I vacancies can form Pb dimer to reduce the Eb. Moreover, as mentioned by authors, the mitigated method of I vacancies by growth of perovskite with I rich condition will lead to enhance the density of I interstitials. Therefore, can we just use some additives, for example Lewis base, to coordinate with uncoordinated Pb and eventually enhance the Eb?

Our reply: We thank the reviewer for this insight comment. Indeed, in addition to halide ions of the common X-site doping (such as Br_I and Cl_I), introducing Lewis base additives would also coordinate with uncoordinated Pb and thus is expected to improve the phase stability of FAPbI₃. In the original manuscript, we have studied the effect of Lewis base additives, namely CN_I and SCN_I, on the phase transition properties for the perovskite. Particularly, SCN_I doping has a positive impact on both phase transition kinetics and thermodynamics, and thus the stability of FAPbI₃.

In the updated manuscript, the related statement has been added on page 12:

“Furthermore, the incorporation of certain additives, such as Lewis bases⁵³, to coordinate with the uncoordinated Pb of V_I⁺ would also suppress the defect-accelerated phase propagation behavior.”

(3) Interestingly, authors demonstrated that inclusion of Ca in perovskite results in disrupting the stability of the perovskite structure. However, the experiment results from literature [Nature Materials 21, 1396–1402 (2022)] have showed that the dopant of Ca contribute to enhancing the stability of perovskite films and photovoltaic performance of devices, which seems to be contradictory with the conclusion in this work. Can the author explain it?

Our reply: Thanks for sharing the relevant literature. Our results show that Ca doping kinetically suppresses phase transition in FAPbI₃, while slightly destabilize α -FAPbI₃

in thermodynamics. Indeed, *Nat. Mater.* **21**, 1396–1402 (2022) reported that Ca doping can suppress the migration of iodine ion in perovskite. On the other hand, it is also reported that Ca doping promotes the formation of a non-perovskite δ -FAPbI₃ during the initial crystallization, especially at the dopant concentration of 5%. This can be attributed to the thermodynamic destabilization induced by Ca doping in the α -phase of FAPbI₃ perovskite based on our DFT predictions, as shown in **Figure 4**.

Accordingly, we cited this important experimental report in the updated manuscript as ref. 56, and updated the related statement on page 15:

“In agreement with our predictions, experimental reports have indicated that Ca doping promotes the formation of a non-perovskite δ -FAPbI₃ during the initial crystallization at the dopant concentrations of 5%⁵⁶.”

(4) Although the phase transition barriers and the phase transition thermodynamics ΔE of pristine and doped FAPbI₃ have been summarized in Figure 3. But the detailed evolution of the lattice parameters, volume, bandgap, and the potential energy as a function of reaction coordinate of the α - δ phase transition in Cs doped FAPbI₃, FAPbI₃ with BrI and ClI defect and some typical B-site doped FAPbI₃ still need to be provided.

Our reply: We would like to thank the reviewer for this comment. We noted that during the phase transition, the evolution of the lattice parameters and volume of doped perovskite systems (at theoretical concentration of 5.56%) exhibits a negligible difference compared with that of the pristine system, as shown in the figure below for the examples of Cs- and Br-doped systems.

Figure 1. Evolution of volume as a function of reaction coordinate of the α - δ phase transition in FAPbI₃ and that doped with Be and Cs, respectively.

In the updated manuscript, as the reviewer suggested, we provided the evolution of bandgap, and the potential energy as a function of the reaction coordinate of the α - δ phase transition in Cs-, Br-, Cl-, La-, and Ce-doped systems in **Figure S7**:

Figure S7. The evolution of the (a) bandgap and (b) potential energy as a function of the reaction coordinate of the α - δ phase transition in the representative Cs-, Br-, Cl-, La-, and Ce-doped FAPbI₃.

and the related statement has been added on page 15:

“The evolution of the bandgap and potential energy as a function of the reaction coordinate of the α - δ phase transition in the representative Cs-, Br-, Cl-, La-, and Ce-doped systems are shown in **Figure S7**.”

(5) The authors have shown the phase transition thermodynamics of the B- and X-site doped FAPbI₃ as a function of the octahedral factor. For me, the linear relation between octahedral factor and phase transition thermodynamics ΔE is just too good to be true.

Our reply: As the reviewer suggested, we have thoroughly double-checked the calculations, especially the phase transition thermodynamics of B- and X-site doped FAPbI₃, and confirmed the results as shown in **Figure 5b**. We noted that while some dopant species exhibit deviation (such as Ca²⁺, Sn²⁺, Ba²⁺, and CN⁻), a roughly volcano-shaped phase transition thermodynamics of the doping systems was observed depending on the octahedral factor, which is an important factor determining the coordination number of the cation and anion in stable perovskite crystal structures.

In the revised manuscript, in the caption of **Figure. 5b**, we have added “The dashed

lines are guides to the eye”.

(6) The calculations in this work demonstrating the synergic stabilizing effects of the exemplary Cs-La and Cs-Ce doped systems. However, why not provide the experimental results to further support the effectiveness of A-B mixed dopant on phase stability of FAPbI₃. We believe that the author should provide the corresponding experimental results to determine the reliability of the calculation results, which also meets the requirements of the journal.

Our reply: Inspired by the reviewer’s suggestion, to verify our theoretical results and examine the effectiveness of A-B mixed doping in improving the phase stability of FAPbI₃ perovskite, we have pursued the experiments of Cs-Eu mixed doping, which is predicted to significantly stabilize the FAPbI₃ perovskite as shown in **Figure 4**. To this end, we grew the single crystals of Cs-doped and Cs-Eu mixed-doped FAPbI₃ perovskite, as shown in **Figure S9**. Notably, the Cs-doped FAPbI₃ is predicted to enhance only the thermodynamical stability, and Eu plays a significant role in kinetically stabilizing the α -phase of FAPbI₃ perovskite. Energy-dispersive X-ray spectroscopy (EDS) measurement results on the Cs-Eu mixed-doped FAPbI₃ single crystal confirm the uniform distribution of Eu element in the single-crystal perovskite sample (**Figure S10**). Indeed, the optical bandgap derived from the Tauc plots using the measured absorption spectra (**Figures 6c 6d**) shows a value of 1.50-1.52 eV for Cs-Eu mixed-doped FAPbI₃ crystal, which agrees with 1.53 eV value predicted by our HSE-SOC calculations.

To assess the phase stability in the Cs-Eu mixed doped α -FAPbI₃ perovskite, we exposed the samples to air for 30 days. The two as-grown single-crystal samples (namely, Cs-doped and Cs-Eu mixed-doped FAPbI₃) initially exhibit black perovskite phases (α -phase) with high purity, as confirmed by the relevant X-ray diffraction (XRD) patterns (see **Figures 6a 6b**). After 30-day exposure, the Cs-doped sample visibly degraded to the characteristic yellow δ -FAPbI₃ phase. In contrast, the Cs-Eu mixed-doped perovskites largely maintained the black α -FAPbI₃ phase without significant changes in colour. This trend was also evident in the XRD results (**Figures 6a 6b**) of the samples measured after the 30-day and 45-day exposures and the corresponding absorption spectra (**Figure 6e**), where the average absorbances at 790 nm after 30-day exposure are dropped by 28.5% and 5.2% for Cs-doped and Cs-Eu mixed-doped (0.5% Eu doping) FAPbI₃ perovskite single crystals, respectively.

Moreover, we investigated the carrier mobility and the trap density, which are the key semiconducting parameters for optoelectronic applications, in single-dopant and mixed-dopant single-crystal perovskite samples, based on the Space-Charge-Limited Current (SCLC) method. Previous experimental studies have reported that α - δ phase transition in FAPbI₃ leads to a significant degradation in carrier transport property. Indeed, our DFT calculations of the effective masses of electrons and holes in the yellow δ -phase ($m_e^* = 0.945 m_0$; $m_h^* = -1.137 m_0$), are around 4.3 and 5.7 times

higher than those in black α phases of FAPbI₃ ($m_e^* = 0.218 m_0$; $m_h^* = -0.199 m_0$), respectively, also as explicitly evidenced by the substantially flatter band edges in δ -FAPbI₃. (**Figures S1c and S1e**). **Figures 6f-i** show the current-voltage (I - V) characteristics for the Cs-doped and Cs-Eu mixed-doped FAPbI₃ single crystals in both fresh statues and after 30 days. Notably, compared with the Cs-doped counterpart ($n_t \sim 6.0 \times 10^9 \text{ cm}^{-3}$; $\mu \sim 121.0 \text{ cm}^2 \text{ V}^{-1} \text{ s}^{-1}$), a lower trap density and higher carrier mobility ($n_t \sim 3.1 \times 10^9 \text{ cm}^{-3}$; $\mu \sim 217.9 \text{ cm}^2 \text{ V}^{-1} \text{ s}^{-1}$) were obtained for the freshly prepared Cs-Eu mixed-doped perovskite single crystal, confirming the promoting effect of Cs-Eu mixed doping on carrier transport of FAPbI₃ perovskites. After 30-day exposure, we observed a more significant decline in carrier mobility in Cs-doped sample ($\Delta\mu \sim 96.5 \text{ cm}^2 \text{ V}^{-1} \text{ s}^{-1}$) as compared to Cs-Eu mixed-doped samples ($\Delta\mu \sim 16.3 \text{ cm}^2 \text{ V}^{-1} \text{ s}^{-1}$). The observations agree well with our theoretical predictions, underscoring high effectiveness of the A-B mixed composition engineering in stabilizing FAPbI₃ perovskite.

Therefore, in the revised manuscript, we have added our experimental results in **Figure 6**:

Figure 6. X-ray diffraction patterns of the (a) Cs-doped and (b) Cs-Eu mixed-cation-doped FAPbI₃ perovskite single crystals, before and after 30-day and 45-day air exposures. (c) UV-Vis-NIR absorption spectra of a series of freshly prepared perovskite single crystals. (d) The optical bandgap derived from the Tauc plots (based

on the data in **c**). **(e)** UV–Vis–NIR absorption spectra of the Cs-doped and Cs-Eu mixed-cation-doped samples with the Eu doping concentrations of 0.25%, 0.50%, and 1.00% before and after 30-day exposure. Current–voltage (I - V) curves and relevant typical SCLC analyses of **(f,g)** Cs single-doped and **(h,i)** Cs-Eu mixed-doped FAPbI₃ perovskite single crystals, before and after a 30-day air exposure, respectively. The regions are marked for Ohmic ($n = 1$), trap-filled limit ($n > 3$), and Child’s regime ($n = 2$). V_{TFL} is the trap-filled limit voltage. We calculated the carrier mobility (μ) and trap densities (N_t) by fitting the I - V data.

Figure S9:

Figure S9. Photographs of the Cs-doped and Cs-Eu mixed-doped FAPbI₃ samples, before and after a 30-day air exposure.

Figure S10:

Figure S10. EDS atomic composition of (a) Cs- and (b) Cs-Eu mixed-doped FAPbI₃ perovskite single crystal. (c) EDS mapping of Pb, I, Eu in the Cs-Eu mixed-doped FAPbI₃ perovskite single crystal.

We also added the related discussion on pages 19-20:

“We noted that the majority of the proposed A-B mixed doped FAPbI₃ perovskites denoted by the purple stars in **Figure 4** have not yet been reported. To validate our theoretical findings, we pursued experimental studies on Cs-Eu mixed doping in FAPbI₃ as a representative example, which is predicted to significantly stabilize the α -FAPbI₃ - both kinetically and thermodynamically. To this end, we first grew single crystals of Cs-doped and Cs-Eu mixed-doped FAPbI₃ perovskites (**Figure S9**), and the related synthetic details are shown in the Experimental Method section. Energy-dispersive X-ray spectroscopy (EDS) measurement results on the Cs-Eu mixed-doped FAPbI₃ single crystal confirm the uniform distribution of the Eu element in the

perovskite (**Figure S10**).

To assess the phase stability in the Cs-Eu mixed-doped perovskite single crystal, we then exposed the samples to air for 30 days. Initially, both as-grown samples (namely, Cs-doped and Cs-Eu mixed-doped FAPbI₃) exhibited black perovskite phases (α -phase) with high purity, as confirmed by the relevant X-ray diffraction (XRD) patterns (see **Figures 6a,b**). After the 30-day exposure, the Cs-doped sample visibly degraded to the yellow δ -FAPbI₃ phase (**Figure S10**). In contrast, the Cs-Eu mixed-cation-doped perovskite largely maintained the black α -FAPbI₃ phase without significant changes in colour. This trend was also evident in the XRD spectra of the samples measured after 30-day and 45-day exposures. **Figures 6c-e** show the UV-Vis-NIR absorption spectra of the freshly prepared samples with different concentrations and those after the 30-day exposure. The optical bandgap derived from the Tauc plots exhibited a value of 1.50-1.52 eV for Cs-Eu mixed-doped FAPbI₃ (**Figure 6d**), in agreement with the value of 1.53 eV predicted by our HSE-SOC calculations. Indeed, compared with the Cs-doped FAPbI₃, we note that the bandgap and absorption profile of the Cs-Eu mixed-doped sample showed negligible change after 30-day exposure. Specifically, the average absorbances at 790 nm after 30-day exposure were dropped by 28.5% and 5.2% for Cs-doped and Cs-Eu mixed-doped (0.5% Eu doping) FAPbI₃ perovskite single crystals, respectively (see **Figure 6e**).”

and on page 22:

“We further investigated the carrier mobility and the trap density, which are the key semiconducting parameters for optoelectronic applications, by using the Space-Charge-Limited Current (SCLC) method⁶⁴ based on the hole-only device (see Supplementary Methods). Previous experimental studies⁶⁵ have shown that α - δ phase transition in FAPbI₃ leads to a significant degradation in carrier transport property. Indeed, our DFT calculations of the effective masses of electrons and holes in the yellow δ -phase ($m_e^* = 0.945 m_0$; $m_h^* = -1.137 m_0$), are around 4.3 and 5.7 times higher than those in black α -phases of FAPbI₃ ($m_e^* = 0.218 m_0$; $m_h^* = -0.199 m_0$), respectively, also as explicitly evidenced by the substantially flatter band edges in δ -FAPbI₃ (**Figures S1c and S1e**). **Figures 6f-i** show the current-voltage (I - V) characteristics for the Cs single-doped and Cs-Eu mixed-doped FAPbI₃ single crystals in both fresh statues and after 30 days. The I - V curves can be divided into three regions: the first, second, and third regions stand for the ohmic ($n = 1$), trap-filling ($n > 3$), and child ($n = 2$) regions, respectively, where the trap density (N_t) and carrier mobility (μ) were calculated from the second and third regions (see the details in Supplementary Information). It is noted that, as compared with the Cs-doped sample ($n_t \sim 6.0 \times 10^9 \text{ cm}^{-3}$; $\mu \sim 121.0 \text{ cm}^2 \text{ V}^{-1} \text{ s}^{-1}$), a lower trap density ($n_t \sim 3.1 \times 10^9 \text{ cm}^{-3}$) and a higher hole mobility ($\mu \sim 217.9 \text{ cm}^2 \text{ V}^{-1} \text{ s}^{-1}$) were obtained for the freshly prepared Cs-Eu mixed-doped perovskite single crystal, confirming the promoting effect of Cs-Eu mixed doping on carrier dynamics of FAPbI₃ perovskites. After 30-day exposure, we

observed a significant decline of hole mobility in Cs-doped sample ($\Delta\mu\sim 96.5\text{ cm}^2\text{V}^{-1}\text{s}^{-1}$), as compared to that of Cs-Eu mixed-doped sample ($\Delta\mu\sim 16.3\text{ cm}^2\text{V}^{-1}\text{s}^{-1}$), highlighting the remarkable effectiveness of A-B mixed composition engineering in stabilizing FAPbI₃ perovskite.”

and updated the experimental method in SI:

“

Experimental Section

Crystal growth. All the series of mixed-cation Pb-based perovskite single crystals were grown using the inverse temperature crystallization (ITC) method. Specifically, a stoichiometric molar ratio of CsFA precursor solution (1.0 M) was prepared by dissolving 1.55 g FAI (9.0 mmol), 0.26 g CsI (1.0 mmol), 0.55 g PbBr₂ (1.5 mmol), and 3.85 g PbI₂ (8.5 mmol) compounds in 10 ml γ -butyrolactone (GBL). After stirring at room temperature for 12 h, the mixed solution was filtered through a 0.22 μm polytetrafluoroethylene filter and transferred into glass vials. Thereafter, the glass vials with the filtered solution were placed on the hot plate heated to 80 °C and kept this temperature for 3 h. Then, the temperature was gradually increased from 80 °C to 120 °C at a slow rate of 2 °C per hour. Small crystals would be formed in the crystallizing dish. For Cs/Eu-doped mixed-cation perovskite single crystal, a small amount of EuI₃ was dissolved into the solution of Cs_{0.1}FA_{0.9}Pb(I_{0.9}Br_{0.1})₃ at a specific concentration. The amounts of B-site dopants (Eu: 0.25%, 0.50% and 1.00%) in the perovskite solution are calculated according to the molar ratio with Pb.

Device fabrication. The hole-only device with vertical Au/Perovskite/Au structure was prepared for the SCLC method. An 80-nm gold (Au) anode was deposited on the top surface of the mixed-cation perovskite single crystal by thermal evaporation (AJA ATC-1800-E e-beam thermal evaporator). Then, a 105 nm Au cathode was deposited on the bottom surface of the single-crystal samples. The obtained devices will be used for the SCLC measurements.

Characterizations. The XRD patterns of the perovskite powder (by gridding the single crystals into powder) were measured using on PANalytical Pro Powder Diffractometer with a Cu K α radiation source. The absorption spectrum measurement was taken in an Agilent Cary 5000 UV-vis-NIR spectrophotometer. Energy-dispersive X-ray spectroscopy (EDS) testing were carried out using Zeiss ULTRA plus under the operating voltage of 20 kV. The I - V characteristic curves of perovskite single crystals were measured using a two-terminal probe station (Everbeing Int'l Corp.) and 4200A-SCS Parameter Analyzer.

Supplementary Notes

The details for SCLC analyses

We also performed current–voltage (I – V) measurements to assess the charge transport properties and trap behaviors of both Cs- and Cs-Eu mixed-cation-doped FA-based perovskite single crystals. The space-charge-limited-current (SCLC) analyses were carried out based on the obtained I – V data. As shown in **Figures 6f-i** in the main text, we can clearly find that three regions were identified in the I – V curves: Ohmic ($n = 1$), trap-filling ($n > 3$), and Child’s ($n = 2$) regions.

At the low applied voltage, an Ohmic region is confirmed. Following the first region, it exhibits a rapid nonlinear rise in current starting at the point of trap-filled limit (TFL) voltage V_{TFL} , indicating the transition into the trap-filling region. In the second region, all the trap states are expected to be filled by the injected carriers. The trap density (N_t) can be evaluated by the following equation:

$$N_t = \frac{2V_{TFL}\epsilon\epsilon_0}{ed^2}$$

in which ϵ is the relative dielectric constant materials, ϵ_0 is the vacuum permittivity, e is the electronic charge, and d represents the thickness of the single crystals.

The carrier mobility (μ) could be extracted from the third region (Child’s region) that shows trap-free characteristic at high bias, following the Mott–Gurney’s SCLC theory expressed by the following equation:

$$\mu = \frac{8d^3}{9\epsilon\epsilon_0} \frac{\partial J}{\partial(V^2)}$$

where J is current density and V is the corresponding applied voltage.”

(7) I think the doping concentration will also play a crucial role for the phase stability. But there is no detailed discussion for the concentration in this work.

Our reply: As the reviewer suggested, we experimentally investigated the phase stability improvement of the Cs-Eu mixed-doped perovskite samples as a function of Eu doping concentration, as shown in **Figure 6c-e**. With increasing the concentration of Eu doping, the degradation of single-crystal perovskite optical properties would be effectively suppressed (**Figure 6e**). Notably, the optimal concentration of Eu doping is 0.5%, where the average absorbances at 790 nm after 30-day exposure are dropped by only 5.2% for Cs-Eu mixed-doped FAPbI₃ perovskite sample as compared to 28.5% for the Cs-doped counterpart.

Therefore, in the revised manuscript, we have added the related experimental data in **Figures 6c-e**:

Figure 6. (c) UV–Vis–NIR absorption spectra of a series of freshly prepared perovskite single crystals. (d) The optical bandgap derived from the Tauc plots (based on the data in c). (e) UV–Vis–NIR absorption spectra of the Cs-doped and Cs-Eu mixed-cation-doped samples with the Eu doping concentrations of 0.25%, 0.50%, and 1.00% before and after 30-day exposure.

and provide the related statement on pages 21-22:

“Moreover, we investigated the phase stability improvement of the Cs-Eu mixed samples as a function of Eu dopant concentration. With increasing the concentration of Eu doping, the degradation of single-crystal perovskite optical properties was effectively suppressed (**Figure 6e**). The optimal concentration of Eu doping was tested to be 0.5%.”

Reviewer #2

Comment: The article entitled “Toward stabilization of formamidinium lead iodide perovskites by defect control and composition engineering” addresses the challenge of phase instability in FAPbI₃-based perovskite solar cells. The authors apply computational simulations to investigate the mechanism behind the undesired α - δ phase transition of FAPbI₃. Iodide vacancies and interstitials are found to significantly impact the transition kinetics by inducing covalency during transition states. The study also highlights the detrimental effects of atmospheric moisture and oxygen on the perovskite phase. Furthermore, the authors propose A-B mixed doping could be potentially an effective strategy for stabilizing the desired α -phase.

The systematic study on different defects and dopings in this work demonstrates a thorough investigation of the factors influencing phase instability in FAPbI₃-based perovskite solar cells. By elucidating the role of these defects, the authors provide important guidance for defect control strategies and potential avenues for improving the stability of FAPbI₃ perovskite materials. As such, I am supportive of publication and would suggest a few points that the authors could consider to discuss in the next revision.

Our reply: We would like to thank the reviewer for the affirmation and positive comments on our manuscript, which is of great encouragement for us. For the additional constructive comments and suggestions, we have detailed our responses as shown below.

(1) The results in Fig. 3 suggest that B-site engineering significantly influences the kinetics of the phase transition, while A-site engineering primarily alters the thermodynamics. The study further demonstrates the synergistic stabilizing effects of Cs-La and Cs-Ce mixed doping, leading to a highly stable FAPbI₃ perovskite. However, it is recommended that the authors include information on the optoelectronic properties of these mixed-doped perovskites. Considering the potential impact of B-site doping on the electronic properties, it is essential to evaluate the suitability of these materials for photovoltaic applications. The inclusion of optoelectronic property data would further enhance the significance and applicability of the study.

Our reply: As the reviewer suggested, we studied the optoelectronic bandgaps of the A-B mixed-doped systems (namely, Cs-La, Cs-Ce, Cs-Nd, Cs-Eu, and Cs-Sm), which are shown to effectively enhance phase stability of FAPbI₃ perovskites, based on the HSE-SOC scheme, as shown in **Figure S8**. The calculated bandgap of Cs-Eu mixed-doped FAPbI₃ is 1.53 eV, consistent with our experimental value of 1.50-1.52 eV (**Figures 6c 6d**). Moreover, we experimentally investigated the carrier mobility and the trap density, which are the key semiconducting parameters for optoelectronic applications, in both Cs-doped and Cs-Eu mixed-doped FAPbI₃ perovskite single crystals, based on Space-Charge-Limited Current (SCLC) technique. The findings reveal that Cs-Eu mixed-doped FAPbI₃, a theoretically superior doping candidate,

outperforms its Cs-only counterpart by exhibiting better carrier transport (much lower trap density and higher carrier mobility) and significantly improved phase stability (Figures 6i-f).

Therefore, in the revised manuscript, we have incorporated the experimental results in Figure 6:

Figure 6. X-ray diffraction patterns of the (a) Cs-doped and (b) Cs-Eu mixed-cation-doped FAPbI₃ perovskite single crystals, before and after 30-day and 45-day air exposures. (c) UV-Vis-NIR absorption spectra of a series of freshly prepared perovskite single crystals. (d) The optical bandgap derived from the Tauc plots (based on the data in c). (e) UV-Vis-NIR absorption spectra of the Cs-doped and Cs-Eu mixed-cation-doped samples with the Eu doping concentrations of 0.25%, 0.50%, and 1.00% before and after 30-day exposure. Current-voltage (*I-V*) curves and relevant typical SCLC analyses of (f,g) Cs single-doped and (h,i) Cs-Eu mixed-doped FAPbI₃ perovskite single crystals, before and after a 30-day air exposure, respectively. The regions are marked for Ohmic ($n=1$), trap-filled limit ($n>3$), and Child's regime ($n=2$). V_{TFL} is the trap-filled limit voltage. We calculated the carrier mobility (μ) and trap densities (N_t) by fitting the *I-V* data.

and **Figure S8**:

Figure S8. Calculated electronic bandgaps for pristine and A-B mixed-doped (namely, Cs-La, Cs-Ce, Cs-Nd, Cs-Eu, and Cs-Sm) FAPbI₃ based on HSE-SOC scheme. The corresponding photovoltaic efficiency limits were presented based on the detailed balance limit theory.

and updated the related statement on page 18:

“Promisingly, based on Shockley–Queisser (SQ) limit⁶³, these doped perovskites still maintain the superior electronic bandgaps, which has direct implications for the observed efficiency in the conversion of solar light (**Figure S8**).”

and on page 20:

“The optical bandgap derived from the Tauc plots exhibited a value of 1.50-1.52 eV for Cs-Eu mixed-doped FAPbI₃ (**Figure 6d**), in agreement with the value of 1.53 eV predicted by our HSE-SOC calculations.”

and on page 21:

“We further investigated the carrier mobility and the trap density, which are the key semiconducting parameters for optoelectronic applications, by using the Space-Charge-Limited Current (SCLC) method⁶⁴ based on the hole-only device (see Supplementary Methods). Previous experimental studies⁶⁵ have shown that α - δ phase transition in FAPbI₃ leads to a significant degradation in carrier transport property. Indeed, our DFT calculations of the effective masses of electrons and holes in the yellow δ -phase ($m_e^* = 0.945 m_0$; $m_h^* = -1.137 m_0$), are around 4.3 and 5.7 times higher than those in black α -phases of FAPbI₃ ($m_e^* = 0.218 m_0$; $m_h^* = -0.199 m_0$), respectively, also as explicitly evidenced by the substantially flatter band edges in δ -FAPbI₃ (**Figures S1c**

and S1e). Figures 6f-i show the current-voltage (I - V) characteristics for the Cs single-doped and Cs-Eu mixed-doped FAPbI₃ single crystals in both fresh states and after 30 days. The I - V curves can be divided into three regions: the first, second, and third regions stand for the ohmic ($n = 1$), trap-filling ($n > 3$), and child ($n = 2$) regions, respectively, where the trap density (N_t) and carrier mobility (μ) were calculated from the second and third regions (see the details in Supplementary Information). It is noted that, as compared with the Cs-doped sample ($n_t \sim 6.0 \times 10^9 \text{ cm}^{-3}$; $\mu \sim 121.0 \text{ cm}^2 \text{ V}^{-1} \text{ s}^{-1}$), a lower trap density ($n_t \sim 3.1 \times 10^9 \text{ cm}^{-3}$) and a higher hole mobility ($\mu \sim 217.9 \text{ cm}^2 \text{ V}^{-1} \text{ s}^{-1}$) were obtained for the freshly prepared Cs-Eu mixed-doped perovskite single crystal, confirming the promoting effect of Cs-Eu mixed doping on carrier dynamics of FAPbI₃ perovskites. After 30-day exposure, we observed a significant decline of hole mobility in Cs-doped sample ($\Delta\mu \sim 96.5 \text{ cm}^2 \text{ V}^{-1} \text{ s}^{-1}$), as compared to that of Cs-Eu mixed-doped sample ($\Delta\mu \sim 16.3 \text{ cm}^2 \text{ V}^{-1} \text{ s}^{-1}$), highlighting the remarkable effectiveness of A-B mixed composition engineering in stabilizing FAPbI₃ perovskite.”

(2) Further to the first question, I wonder if the authors considered different A-B mixed doping configurations in their study. Specifically, did they explore multiple combinations of A-site and B-site dopants? Moreover, it would be interesting to know if E_b and $\Delta E_{\delta-\alpha}$ vary across these different configurations.

Our reply: According to the reviewer’s suggestion, we further explored the effects of Cs-Nd, Cs-Eu, Cs-Sm mix-doping on the phase stability of FAPbI₃, and the results are shown in **Figure 4**. These results further affirm the efficiency of our proposed A-B mixed doping strategy, especially with rare-earth elements as the B component.

Figure 4. Kinetic α - δ phase transition barrier E_b and thermodynamic phase energy difference $\Delta E_{\delta-\alpha}$ of pristine FAPbI₃ and those with dopants and impurity interstitials. B-site cation engineering predominantly impacts the kinetics of the FAPbI₃ phase transition that dedicates the device longevity, as depicted by the green dashed circle (ellipse), whereas A-site doping more effectively alters the phase transition thermodynamics that can impact the perovskite crystallization, as depicted by the red dashed circle (ellipse). Moreover, the yellow region highlights various lanthanide doping and the purple stars stand for A-B mixed doping. **The evolution of the bandgap and potential energy as a function of the reaction coordinate of the α - δ phase transition in the representative Cs-, Br-, Cl-, La-, and Ce-doped systems are shown in Figure S7.**

Moreover, the related discussion was updated on page in the updated manuscript:

“This is confirmed by calculations demonstrating the synergic stabilizing effects of the exemplary Cs-La, Cs-Ce, Cs-Nd, Cs-Eu, and Cs-Sm doped systems (see the purple stars in Figure 4).”

(3) The authors further conducted AI-MD simulations to investigate the mechanism driving the α - δ phase transition of FAPbI₃. However, considering the relatively short simulation time, the results obtained from these simulations may not provide conclusive evidence. Therefore, I would recommend moving this section to the supporting information.

Our reply: Thanks for pointing out this negligence. In the updated manuscript, we prolonged the AIMD simulation time from 10 ps to 30 ps, and moved the related results to the supporting information as **Figure S3**:

Figure S3. Mean square displacement (MSD) of the inorganic skeleton within 30 ps during AIMD simulations at 300 K: (a) for pristine and systems containing intrinsic defects, and (b) for pristine and Ln-doped systems.

Reviewer #3

Comment: In the manuscript, the author presents extensive first-principles atomistic calculations of the phase transition of FAPbI₃ from α -phase to δ -phase. However, despite the author's thorough computation and comparison of activation energy barriers throughout the manuscript, the primary conclusions rely solely on these activation energy barrier calculations. This lack of cross-verification with multiple sets of computational results undermines the solidity of the author's conclusions. For example, on page 10 line 269-272, the author mentions, "Given that V_I⁺ defects play the most significant role in the local kinetic stability of FAPbI₃ with a sharp decline of ~ 0.3 eV/f.u., nucleation for the phase transition is likely to occur around the defect centers due to a much faster nucleation rate." I do not believe that such a conclusion, particularly regarding nucleation, can be drawn solely from an energy perspective. In reality, the nucleation process is a complex phenomenon influenced by multiple factors of both kinetics and thermodynamics. It would require additional verification through dynamic nucleation simulations, such as those conducted using AIMD (Ab Initio Molecular Dynamics) methodology or Classical MD. Performing additional phase transition simulations based on AIMD may offer more dynamic insights and information, potentially addressing the limitations of this manuscript, which heavily relies on DFT static calculations.

Our reply: Firstly, we thank the reviewer for the insightful comments on our work. Actually, it is challenging to simulate the nucleation of α -to- δ phase transition of significant atomic displacement in pure α -FAPbI₃ based on the conventional AIMD calculation, due to limitations of length-scale (namely supercell size) and simulation time-scale (picosecond timescale for AIMD calculation, while typically the α -to- δ phase transition of FAPbI₃ occurs in millisecond-to-hour timescale) (*ACS Energy Lett.* **7**, 1534–1543 (2022)). Nevertheless, to address this problem and assess the effect of defects on the phase transition of FAPbI₃, we constructed a configuration that includes a planar interface between the α -FAPbI₃ (111) and δ -FAPbI₃ (100) phases, as shown in **Figure S4**, which has been reported to be the lowest-energy interface structure between the two phases (*NPG Asia Mater* **13**, 1–8 (2021)) with an in-plane lattice mismatch less than 1%. Total energy calculations show that V_I⁺ energetically prefers to reside more favorably at the interfaces than in the bulk of α -FAPbI₃ and δ -FAPbI₃ by 4.38 and 2.14 eV. Combining with the low diffusion barrier as previously reported (*Nat Commun* **6**, 7497 (2015)), V_I⁺ defects are expected to diffuse and aggregate around the interface between the two phases of FAPbI₃. Subsequently, we performed long-time scale NPT (constant-temperature and constant-pressure) ensemble Machine Learning Molecular Dynamics (MLMD) simulation based on the training data set from on-the-fly hybrid AIMD and MLMD simulations to investigate the behaviour of α -to- δ phase propagation for pristine and V_I⁺-defective systems.

While we did not observe clear phase propagation in the pristine system in the 2 ns MLFF simulation, as shown in **Figures 3a, 3b** in the updated manuscript, it is noted the onset of a PbI₆ octahedron of the interfacial plane transitioning from the corner-sharing

to the face-sharing architecture of δ -FAPbI₃ in the proximity of V_I^+ . This can be regarded as the initiation of α -to- δ phase transition of FAPbI₃. This initiation is also reflected by a larger MSD of the Pb-I inorganic skeleton of the interfacial plane than that of pristine systems over the simulation time, as shown in **Figure 3c**. The α -to- δ phase transition of FAPbI₃ is a typical diffusive transformation (*Matter* **4**, 2627–2629 (2021)), in which a larger MSD of the Pb-I inorganic skeleton in the V_I^+ -containing system indicates a more diffusive host ion behaviour that promotes the local atomic reconstruction on the interfacial plane for phase propagation.

Moreover, to further quantify these visual observations, we compared the radial distribution function (RDF) for Pb-Pb in both pristine and V_I^+ -defective systems, as shown in **Figure 3d**. Given the atomic structure of α -FAPbI₃ and δ -FAPbI₃ in **Figure S1**, the first and third sharp peaks correspond to Pb-Pb interactions along the c -direction and the ab plane in δ -FAPbI₃, and the second sharp peak for that in α -FAPbI₃, respectively. As can be seen, compared with the pristine system, the second peak in the V_I^+ -defective system is broadened toward the first and third ones, accompanied by a slight reduction in the coordination number, $g(r, \text{Pb-Pb})$. This signifies a stronger tendency of the α -to- δ phase transition of FAPbI₃ in the V_I^+ -containing system. The MLMD results support our DFT conclusion that the presence of V_I^+ facilitates the α -to- δ phase transition of FAPbI₃.

In the revised manuscript, we have incorporated the MLMD results in **Figure 3**:

Figure 3. Snapshots captured at 0 and 1.8 ns from the MLMD simulation of α -to- δ phase propagation in the (a) pristine and (b) V_I^+ -defective planar interface models between α -FAPbI₃ (111) and δ -FAPbI₃ (100) at 300 K. (c) Mean square displacement

of the Pb-I inorganic skeleton of the interfacial plane (the region surrounded by the red dashed line in **Figure S4b**) and **(d)** the radial distribution function for Pb-Pb during the 2-ns MLMD simulations in the pristine and V_I^+ -defective systems at 300 K.

and **Figure S4**:

Figure S4. The 312-atom **(a)** and 1248-atom **(b)** interface models between α -FAPbI₃ (111) and δ -FAPbI₃ (100) phases.

and provided more discussion based on the MLMD results on pages 11-12:

“Phase transition typically proceeds through the growth and propagation of the targeted phase nuclei. Given that V_I^+ plays the most significant role in the local kinetic stability of FAPbI₃ with E_b suffering the sharpest decline by ~ 0.3 eV/f.u., we expected that the defects effectively facilitate the δ -phase propagation, responsible for a faster overall phase transition rate in FAPbI₃. To gain dynamic insights into the mechanism of phase transition aided by V_I^+ and verify our DFT results, we proceeded to construct a configuration with planar interfaces between the α -FAPbI₃ (111) and δ -FAPbI₃ (100) phases (**Figure S4b**), and perform 2-ns NpT Machine Learning Molecular Dynamics (MLMD) simulation at 300 K based on the training data set from on-the-fly hybrid AIMD. Energy calculations revealed that V_I^+ energetically prefers to reside at the interface than in the bulk of α -FAPbI₃ and δ -FAPbI₃ by 4.38 and 2.14 eV, respectively. Combining with the low diffusion barrier as previously reported⁵¹, V_I^+ defects are expected to diffuse and aggregate around the interface between the two phases of FAPbI₃.

As shown in **Figures 3a, b**, while no clear phase propagation was observed in the pristine system during the 2-ns MLFF simulation, a sign of notable transformation was evident in the V_1^+ -defective system. Specifically, a PbI_6 octahedron at the interfacial plane around V_1^+ transformed to the face-sharing architecture of δ -FAPbI₃. This is further reflected by the analysis of MSD, as shown in **Figure 3c**. The Pb-I inorganic skeleton on the interfacial plane exhibited a significantly larger MSD compared to that of the pristine systems throughout the simulation time. For a typical diffusive transformation associated with the α - δ phase transition in FAPbI₃⁵², a larger MSD of the Pb-I inorganic skeleton in the V_1^+ system indicates a more diffusive host ion property around the two-phase interface. The increased diffusivity significantly promotes the local atomic rearrangement and reconstruction of the interfacial plane, thereby promoting phase propagation.

Moreover, to quantify these visual observations, we compared the radial distribution function (RDF) for Pb-Pb interaction in both the pristine and V_1^+ -defective systems (**Figure 3d**). For the ground-state atomic structures of α -FAPbI₃ and δ -FAPbI₃ (**Figure S1**), the second sharp peak corresponds to that in α -FAPbI₃, and the first and third sharp peaks stand for Pb-Pb along the c -direction and the ab -plane in δ -FAPbI₃, respectively. As can be seen, compared with the pristine system, the second peak in the V_1^+ system is broadened toward the first and third ones, accompanied by a slight reduction in coordination number. This implies a stronger tendency of the α -to- δ phase transition of FAPbI₃ in the V_1^+ system. Thus, a higher density of V_1^+ defects can result in an increased propagation rate of the undesired α -to- δ phase transition in FAPbI₃. The MLMD simulation provides a dynamic understanding of the DFT results.”

and updated the computational method in SI:

“**Machine Learning Molecular Dynamics.** Machine Learning Molecular Dynamics (MLMD)^{7,8} was performed as implemented in VASP. All the datasets were trained based on the on-the-fly hybrid AIMD and MLMD calculations, in which first-principles *ab initio* calculations will be conducted when local molecular environments are significantly different from those already stored as the training data. In this study, Machine Learning Force Field (MLFF) was trained in an isobaric–isothermic (NpT) ensemble with a time step of 1 fs, using the ($3 \times 3 \times 2$) 216-atom bulk supercells of α -FAPbI₃ and δ -FAPbI₃, 360-atom surface models of α -FAPbI₃ (111) and δ -FAPbI₃ (100) with a vacuum thickness of 15 Å, 312-atom (**Figure S4a**) and 1248-atom (**Figure S4b**) interface models between the α -FAPbI₃ (111) and δ -FAPbI₃ (100) phases, and those with a V_1^+ . This interface model has been reported to exhibit the lowest total energy⁹, and the lattice mismatch was less than 1%. The force field was generated with a cutoff radius of 7 Å for the angular descriptor and a width of 0.5 Å of Gaussian functions for broadening the atomic distributions of the radial descriptor. The MLFF training involves 10-ps NpT simulations at 200 K, 300 K, and 400 K for each configuration. Based on the obtained datasets of MLFF, 2 ns NpT-MLMD simulations with a time

step of 1 fs were run for the 1248-atom interface models of pristine FAPbI₃ and that containing a V_I⁺.”

Additionally, the MSD curve in Figure 2(c) does not reach a linear trend, showing unstable or even decreasing trends. This is likely due to restricted ion movement in confined lattice structure, rendering the MSD calculation data unsuitable for supporting the conclusions presented in page 9 256-258. A near flat MSD curve implies that the particle is essentially confined to lattice sites and undergoes localized vibrations. Based on the data presented in the manuscript so far, the support for the results is not sufficiently solid and comprehensive. Therefore, I recommend that this article is not suitable for publication in its current form.

Our reply: We thank the reviewer for this insightful suggestion. To refine our AIMD results, we have increased the supercell size (from 3 × 3 × 2 supercells with 216 atoms to 3 × 3 × 3 supercells with 324 atoms) and prolonged the AIMD simulation time (from 10 ps to 30 ps). Our refined AIMD results do not alter our conclusions. Nevertheless, considering the limitations involved, in the revised manuscript, we have moved the updated AIMD results to Supporting Information as **Figure S3**:

Figure S3. Mean square displacement (MSD) of the inorganic skeleton within 30 ps during AIMD simulations at 300 K: (a) for pristine and systems containing intrinsic

defects, and **(b)** for pristine and Ln-doped systems.

and updated the related statement on page 8:

“we performed AIMD simulations on pristine α -FAPbI₃ and systems containing typical intrinsic defects at room temperature over a 30-ps period with a time step of 1 fs”

REVIEWERS' COMMENTS

Reviewer #1 (Remarks to the Author):

The authors have addressed all the issues. Especially, the authors have pursued the experiments of Cs-Eu mixed doping. The results not only proved the effectiveness of A-B mixed doping in improving the phase stability of FAPbI₃ perovskite, but also completed the suggested mechanism and enhanced the novelty. Hence, I would like to propose this manuscript for your consideration regarding publication.

Reviewer #2 (Remarks to the Author):

After carefully reviewed the revised version of the manuscript entitled "Toward stabilization of formamidinium lead iodide perovskites by defect control and composition engineering", I find that the manuscript now addresses the concerns raised, and the additional experimental results provided have further strengthened the study. In addition, the authors have included optoelectronic property data for the mixed-doped perovskites in the revised manuscript. This addition enhances the significance and applicability of the study, as it provides important insights into the suitability of these materials for photovoltaic applications. Therefore, I find that the manuscript now meets the standards for publication.

Reviewer #3 (Remarks to the Author):

In the revised manuscript, the author's efforts to supplement the manuscript with experimental and AIMD-related data, along with the emphasis on providing cross-verification, are commendable. The inclusion of additional experimental and AIMD data significantly strengthens the robustness of their findings. I understand that certain technical challenges posed limitations on achieving complete resolutions in some aspects of the study, like fulfilling simulate the nucleation of α -to- δ phase transition. Given these constraints, I accept the current state of the results, recognizing the efforts authors have made to optimize the findings within the existing framework. Looking ahead, I am hopeful that future research will provide opportunities to delve more deeply into the aspects that couldn't be fully addressed in the current study. I suggest the manuscript can be accept in current form.

RESPONSE TO REVIEWERS' COMMENTS

Reviewer #1 (Remarks to the Author):

The authors have addressed all the issues. Especially, the authors have pursued the experiments of Cs-Eu mixed doping. The results not only proved the effectiveness of A-B mixed doping in improving the phase stability of FAPbI₃ perovskite, but also completed the suggested mechanism and enhanced the novelty. Hence, I would like to propose this manuscript for your consideration regarding publication.

Our reply: We greatly appreciate the Reviewer's time and efforts in reviewing our manuscript again. We would like to thank the Reviewer for the affirmation on our work and for recommending publication of our manuscript in Nature Communications.

Reviewer #2 (Remarks to the Author):

After carefully reviewed the revised version of the manuscript entitled "Toward stabilization of formamidinium lead iodide perovskites by defect control and composition engineering", I find that the manuscript now addresses the concerns raised, and the additional experimental results provided have further strengthened the study. In addition, the authors have included optoelectronic property data for the mixed-doped perovskites in the revised manuscript. This addition enhances the significance and applicability of the study, as it provides important insights into the suitability of these materials for photovoltaic applications. Therefore, I find that the manuscript now meets the standards for publication.

Our reply: We are highly encouraged by the Reviewer for his/her thought that our work 'provides important insights into the suitability of these materials for photovoltaic applications'. We would like to thank the Reviewer for the affirmation on our work and for recommending our work to be published in Nature Communications.

Reviewer #3 (Remarks to the Author):

In the revised manuscript, the author's efforts to supplement the manuscript with experimental and AIMD-related data, along with the emphasis on providing cross-verification, are commendable. The inclusion of additional experimental and AIMD data significantly strengthens the robustness of their findings. I understand that certain technical challenges posed limitations on achieving complete resolutions in some aspects of the study, like fulfilling simulate the nucleation of α -to- δ phase transition. Given these constraints, I accept the current state of the results, recognizing the efforts authors have made to optimize the findings within the existing framework. Looking ahead, I am hopeful that future research will provide opportunities to delve more deeply into the aspects that couldn't be fully addressed in the current study. I suggest the manuscript can be accept in current form.

Our reply: We are particularly excited by the Reviewer's suggestion of acceptance of our manuscript. We would like to thank the Reviewer for the affirmation on our work.